# MERLOT:
# Multimodal Neural Script Knowledge Models

Rowan Zellers♠🍷   Ximing Lu♠♡🍷   Jack Hessel♡🍷
Youngjae Yu♡   Jae Sung Park♠   Jize Cao♠♡   Ali Farhadi♠   Yejin Choi♠♡
♠Paul G. Allen School of Computer Science & Engineering, University of Washington
♡Allen Institute for Artificial Intelligence
https://rowanzellers.com/merlot

## Abstract

As humans, we understand events in the visual world contextually, performing multimodal reasoning across time to make inferences about the past, present, and future. We introduce MERLOT, a model that learns multimodal script knowledge by watching millions of YouTube videos with transcribed speech – in an entirely label-free, self-supervised manner. By pretraining with a mix of both frame-level (spatial) and video-level (temporal) objectives, our model not only learns to match images to temporally corresponding words, but also to contextualize what is happening globally over time. As a result, MERLOT exhibits strong out-of-the-box representations of temporal commonsense, and achieves state-of-the-art performance on 12 different video QA datasets when finetuned. It also transfers well to the world of static images, allowing models to reason about the dynamic context behind visual scenes. On Visual Commonsense Reasoning, MERLOT answers questions correctly with 80.6% accuracy, outperforming state-of-the-art models of similar size by over 3%, even those that make heavy use of auxiliary supervised data (like object bounding boxes).

Ablation analyses demonstrate the complementary importance of: 1) training on videos versus static images; 2) scaling the magnitude and diversity of the pretraining video corpus; and 3) using diverse objectives that encourage full-stack multimodal reasoning, from the recognition to cognition level.

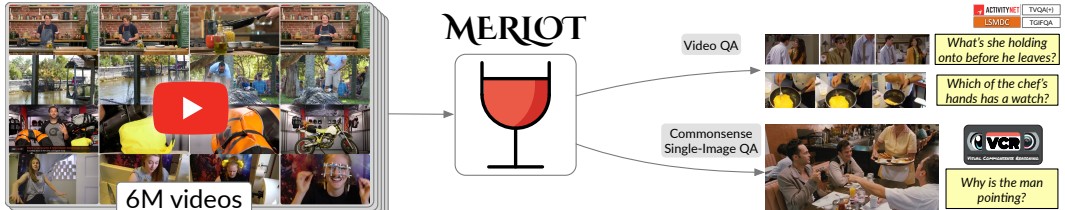

Figure 1: **M**ultimodal **E**vent **R**epresentation **L**earning **O**ver **T**ime. We learn representations of multimodal script knowledge from 6 million YouTube videos. These representations can then be applied to a variety of downstream tasks that require commonsense or temporal visual reasoning.

## 1   Introduction

The human capacity for commonsense reasoning is shaped by how we experience causes and effects over time. Consider the still image of people dining at a restaurant in the bottom right of Figure 1: while a literal, concrete description like "people sitting at a table eating" might be technically correct for the static scene, it doesn't capture the richer temporal, commonsense inferences that are nonetheless obvious: *before* sitting down, the people had to meet up, agree where to go, and enter the

---

🍷: Equal contribution.
35th Conference on Neural Information Processing Systems (NeurIPS 2021).

restaurant; *at present*, the man is pointing because the server just came to the table, and she might want to know whose food is whose; and *after*, it is likely the server will return to the kitchen to help another table.

Teaching machines this type of *script knowledge* [95] is a significant challenge in no small part because enumerating all facts, inferences, and counterfactuals is prohibitive. As a result, the highest performing models on vision-and-language tasks, including Visual Commonsense Reasoning (VCR) (where Figure 1's scene originates from), learn about the visual world exclusively through static images paired with literal captions [108, 22, 69, 75, 119, 36]. Though some captions might hint at the past and future, it is not obvious that even training on, e.g., 400M literal image/text pairs [89] will result in models capable of temporal reasoning.

In this paper, we introduce MERLOT, short for **M**ultimodal **E**vent **R**epresentation **L**earning **O**ver **T**ime. MERLOT is a model that learns commonsense representations of multimodal events by self-supervised pretraining over 6M unlabelled YouTube videos. With the goal of learning multimodal reasoning capacity beyond static images/literal captions, we train MERLOT to **a)** match individual video frames with contextualized representations of the associated transcripts, and to **b)**, contextualize those frame-level representations over time by "unmasking" distant word-level corruptions [27] and reordering scrambled video frames.

We validate our model on a diverse suite of video tasks, requiring both recognition- and cognition-level reasoning across long and short timescales; when finetuned, MERLOT achieves a new state-of-the-art on 12 such tasks. Additionally, we show that our script-knowledge representations transfer to the single image domain. On Visual Commonsense Reasoning (VCR; [123]), our model achieves particularly strong performance, outperforming models that require heavy visual supervision (in the form of object detection bounding boxes, or images paired with pristine captions).

Beyond finetuning, we show both quantitatively and qualitatively that MERLOT has a strong out-of-the-box understanding of everyday events and situations. Given a scrambled visual story, [50, 2], MERLOT can sort image sequences to match captions which tell a globally coherent narrative. Despite considerable domain shift from videos to static images, MERLOT outperforms strong baselines like CLIP [89] and UNITER [22], which independently match images to text and thus cannot reason over long-term contexts as effectively. This capacity for temporal coherence emerges during pretraining: analysis of MERLOT's attention patterns (Figure 11) show that regions attend to captions that are distant in time (and vice versa), allowing it perform cross-modal coreference to piece together a holistic view of situations.

Finally, ablations of MERLOT show that 1) pretraining works better when we train on videos rather than still images, aided crucially by our strategy of corrupting highly visual words in the masked language modeling task, 2) using a diverse set of videos covering many aspects of everyday situations improves downstream performance compared to curated instructional video corpora [107, 80] which both cover a smaller slice of the visual world (confirming hypotheses from past work [47]); and 3) MERLOT's performance does not saturate even after many epochs of training on the pretraining corpus we curated, YT-Temporal-180M, as it continues to improve performance simply with more pretraining. The combination of these results suggests that learning full-stack visual reasoning and multimodal world knowledge from video data is a promising path forward for future research.

In summary, our main contributions are:

1. MERLOT a performant end-to-end vision and language model, that learns powerful multimodal world representations from videos and their transcripts – using no labeled data.
2. YT-Temporal-180M, a diverse corpus of frames/ASR derived from a filtered set of 6M diverse YouTube videos, which we show greatly aids performance, and
3. A set of experiments/ablations demonstrating the strong performance of MERLOT on a set of 14 tasks, spanning finetuning and zero-shot transfer, and images and videos.

At `rowanzellers.com/merlot`, we have released code, data, and models for public research use.

## 2 Related Work

### 2.1 Joint representations of written text and images

There is a long history of work on learning joint text-image representations [14]. Recently, several papers have proposed "Visual BERT" models [108, 22, 8, 69, 75, 119, 36], trained on image captioning datasets such as MSCOCO [71]. In general, features are extracted using Anderson et al. [10]'s frozen object detector, which was originally trained on Visual Genome [60]. Some exceptions are Zhang et al. [125], who use an even larger object detector trained on more labeled data; Kim et al. [57], who use an ImageNet-pretrained backbone [26], and Shen et al. [100], who study a CLIP backbone [89] pretrained on web image-caption pairs.

Overall, these approaches all learn visual representations of static images, and rely on significant human annotation in doing so (e.g. through literal image descriptions). Instead, our approach learns *dynamic* visual representations purely from videos – their frames, and a transcript of what is said – thus using no human annotation.

### 2.2 Learning from videos, with automatic speech recognition (ASR) transcripts

Prior works have used web videos with ASR to build weakly-supervised object detectors [87], action detectors/classifiers [120, 6, 62, 84], instruction aligners [77, 5, 19], video captioners [96, 46, 86, 101], and visual reference resolvers [49]. Of late, works have sought to learn multimodal representations transferable to many tasks from uncurated sets of (usually how-to) videos [80, 106, 107, 81, 127, 9, 7, 4]; generally these are applied to video understanding tasks like activity recognition. One challenge is designing an appropriate objective for learning video-level representations. Lei et al. [67]'s ClipBERT model learns vision-language representations from image captions, which more literally describe image content versus the longer ASR transcripts we consider. Tang et al. [109] use a pretrained dense image captioner [59] to provide auxiliary labels for web how-to videos. Both approaches use (supervised) ResNets pretrained on ImageNet [43] as their visual backbones. MERLOT is trained using a combination of objectives requiring no manual supervision; it nonetheless outperforms both prior approaches on downstream tasks.

### 2.3 Temporal ordering and forecasting

There has been a large body of work on analyzing 'what happens next' in videos [58]. Some modeling choices include using pixels [34, 113], graphs [11], euclidean distance using sensors [3], or studying cycle consistency across time [32]. In addition to extrapolation, past work has studied deshuffling objectives in videos [82, 115], though this has mostly been limited to the visual modality. In contrast to these papers, our goal is learning *multimodal* script knowledge representations: using both language and vision as complementary views into the world, instead of just tracking what changes on-screen.

## 3 MERLOT: Multimodal Event Representation Learning Over Time

We now present our unified model for learning script knowledge through web videos; including our pretraining dataset, architecture, and objectives.

### 3.1 YT-Temporal-180M

We collect YT-Temporal-180M, a dataset for learning multimodal script knowledge, derived from 6 million public YouTube videos. Our YT-Temporal-180M intentionally spans many domains, datasets, and topics. We began with 27 million candidate video IDs (which we then filtered), including instructional videos from HowTo100M [80], lifestyle vlogs of everyday events from the VLOG dataset [35], and YouTube's auto-suggested videos for popular topics like 'science' or 'home improvement.' Our intent (in making the corpus as diverse as possible) was to encourage the model to learn about a broad range of objects, actions, and scenes [47]: we will later show through an ablation that limiting our pretraining to only instructional videos indeed hurts performance downstream.

We filtered videos using the YouTube API, which provides access to videos themselves, their ASR track (automatically transcribed speech tokens), and other metadata. We discard videos 1) without

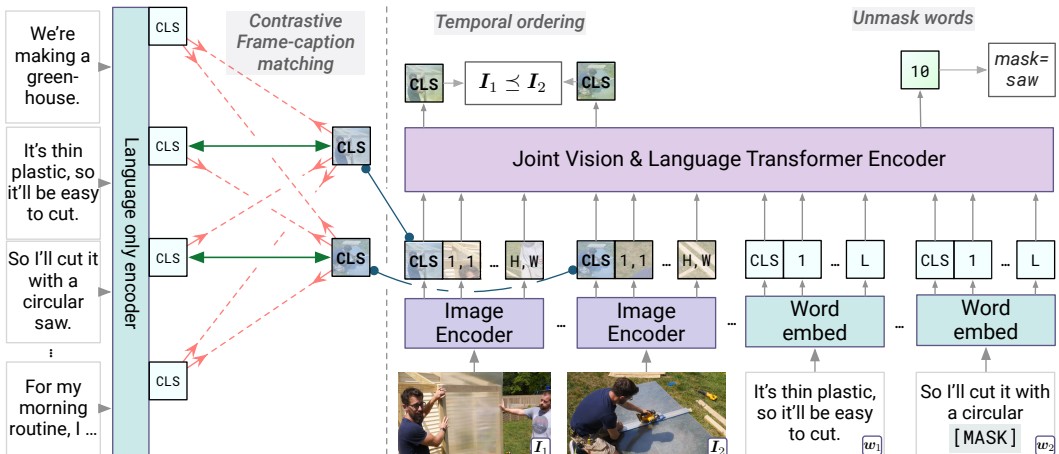

Figure 2: **Left**: $\mathbb{MERLOT}$ learns to match contextualized captions with their corresponding video frames. **Right**: the same image encoding is provided, along with (masked) word embeddings, into a joint vision-language Transformer model; it then unmasks ground words (like 'saw' in this example) and puts scrambled video frames into the correct order.

an English ASR track; 2) that are over 20 minutes long; 3) that belong to visually "ungrounded" categories like video game commentaries; and 4) that have thumbnails unlikely to contain objects, according to a lightweight image classifier. We add punctuation to the ASR by applying a sequence-to-sequence model trained to add punctuation to sentences/paragraphs from news articles. Full details of the scraping and filtering are in Appendix A.

Each video $\mathcal{V}$ might contain thousands of frames. In this work, we represent a video $\mathcal{V}$ as a sequence of consecutive **video segments** $\{s_t\}$. Each segment $s_t$ consists of:

**a.** an image frame $I_t$, extracted from the middle timestep of the segment,

**b.** the words $w_t$ spoken during the segment, with a total length of $L$ tokens.

To split the videos into segments, we byte-pair-encode (BPE; [97, 88]) each video transcript and align tokens with YouTube's word-level timestamps. This enables us to split the videos into segments of $L=32$ BPE tokens each (Appendix A.4); our final dataset has 180 million segments of this form.

## 3.2 $\mathbb{MERLOT}$ **Architecture**

A diagram of $\mathbb{MERLOT}$ is given in Figure 2. $\mathbb{MERLOT}$ takes a sequence of video frames $\{s_t\}$ as input. We encode each frame $I_t$ using an image encoder, embed the words $w_t$ using a learned embedding, and jointly encode both using a Transformer [112]. After pretraining, the architecture can be applied to a variety of vision-and-language tasks with minimal modification. For video QA, for example, we pass several video frames to the image encoder, the question to the text encoder, and extract a single vector representation from the CLS token position. For each task, we learn a lightweight classification head mapping from this hidden state to the task's label space; specific modeling/optimization details are given in Appendix E.2.

**Image encoder.** We train our image encoder end-to-end, alongside the rest of the model, from random initialization (thus without learning from supervised data). While most performant vision-and-language models pre-extract features from a (supervised) object detector [108, 69, 75, 22, 68], for the sake of pre-training efficiency we use a grid-based hybrid ResNet/Vision Transformer.[1]

Specifically: our encoder uses a ResNet-50 backbone, followed by a 12-layer, 768-dimensional Vision Transformer [43, 112, 31]. We made additional modifications that improve efficiency, including: 1) we trained on smaller, widescreen images of size 192x352 (because most YouTube videos are

---

[1]Standard object detectors have expensive operations for proposing regions, and extracting features from those regions (RoI-pooling); our grid approach avoids these. Recent work has proposed using 'grid features' broadly [53], yet on tasks like VCR these approaches have so far underperformed the more expensive object detector backbones [123]; our results suggest that 'grid features' can perform well broadly.

widescreen) using a patch size of 16x16 pixels; 2) we mirror [31]'s alterations of removing the C5 block in ResNet-50; and 3) we save compute further by average-pooling the final-layer region cells using a kernel size of $2 \times 2$. With these modifications, our image encoder requires 40 gigaFLOPs for a forward pass, which is 2% of the 2 teraFLOPs required for the Faster-RCNN.

In summary: given an image of size $W \times H$, the image encoder will output a $W/32 \times H/32$ feature map, along with two CLS hidden states: one for pooling a global representation of the image, and another for pretraining (Task **1**.).

**Joint Vision-Language Encoder.** The joint encoder is a 12-layer, 768-dimensional Transformer [112], mirroring the RoBERTa base architecture [72]; we initialize it with pretrained RoBERTa weights. To compute joint representations, we first embed the tokens $\{w_t\}$ via lookup, and then add position embeddings to both language and vision components (i.e., $\{I_t\}$). The position embeddings differ between different segments, so as to distinguish between images and captions at different timesteps. Finally, we pass the independent visual and textual feature maps to our joint encoder.

The tokens $w_t$ in each segment begin with a CLS token; recall that the feature maps for each frame $I_t$ start with one as well. At those positions, we will later pool final-layer hidden-state representations, for use in pretraining along with downstream tasks.

### 3.3 Pretraining Tasks and Objectives

We use the following three objectives to pretrain $\mathrm{MERLOT}$, that cover 'full-stack' visual reasoning – from recognition subtasks (like object detection) that operate at the frame level, to more 'cognitive' tasks that operate at the video level.

**1**. **Contrastive frame-transcript matching** [126, 89]. We want to ensure that the underlying image encoder produces helpful image representations. Thus, we use the video transcript to compute a 'language-only' representation of each video segment; and use a contrastive loss to maximize its similarity to corresponding representations from the image encoder.[2]

Unlike what is the case for many image captions, the words $w_t$ in each segment are often not sufficient to describe the gist of $I_t$, or even what the key objects might be – for that, video-level contextualization is often required. We thus pass the entire transcript into the language-only encoder, which then extracts hidden states for each segment at the segment-level CLS tokens.

Given matching representations for each frame $I_t$ and caption $w_t$ as positive examples, the negative examples come from all other frame-caption pairs in the batch – whether or not they come from the same video. We project both of these representations into a size-768 hidden state which is then unit-L2-normalized, and compute an all-pairs dot-product between all image and text representations. We divide these logits by a temperature of $\tau = 0.05$, and then apply a pairwise cross entropy loss to encourage matching captions and frames.

**2**. **(Attention) Masked Language Modeling** When providing words into the joint vision-and-language encoder, we randomly replace 20% with a MASK token, a random word, or the same word; $\mathrm{MERLOT}$ must then reconstruct the correct word with a cross-entropy loss, following [27].

This approach is commonly used by 'visual BERT' models in the image captioning domain, where captions are concise, and thus the identity of masked concrete words is difficult for models to recover given language context alone. However, we observed qualitatively that videos break these assumptions: people tend to ramble, and often mention key objects multiple times. Thus, applying vanilla BERT-style masking often causes ungrounded fillers like 'umm' or 'yeah' to get masked, while the (repeated) names of important objects are often partially masked, penalizing the learning of multimodal representations.

We introduce a simple solution to this problem, that we call **attention masking**: we use attention weights from a language-only transformer (introduced in the previous objective) as a heuristic for which words are grounded. 50% of the time, we mask out a random token; the other 50% of the time, we mask out one of the top 20% most-attended-to-tokens. We then apply SpanBERT masking [54], randomly corrupting the following or preceding tokens with an average length of 0.5 tokens in each direction; this makes it harder for models to over-rely on BPE artifacts. We show in ablations that this improves performance.

---

[2]To save memory, our 'language-only encoder' for this subtask shares parameters with the joint vision-and-language encoder.

|  | Q→A | QA→R | Q→AR |
|---|---|---|---|
| ViLBERT [75] | 73.3 | 74.6 | 54.8 |
| Unicoder-VL [68] | 73.4 | 74.4 | 54.9 |
| VLBERT [69] | 73.8 | 74.4 | 55.2 |
| UNITER [22] | 75.0 | 77.2 | 58.2 |
| VILLA [36] | 76.4 | 79.1 | 60.6 |
| ERNIE-ViL [119] | 77.0 | 80.3 | 62.1 |
| MERLOT (base-sized) | **80.6** | **80.4** | **65.1** |

Table 1: Results on VCR [123]. We compare against SOTA models of the same 'base' size as ours (12-layer vision-and-language Transformers). MERLOT performs best on all metrics.

|  | Spearman (↑) | Pairwise acc (↑) | Distance (↓) |
|---|---|---|---|
| CLIP [89] | .609 | 78.7 | .638 |
| UNITER [22] | .545 | 75.2 | .745 |
| MERLOT | **.733** | **84.5** | **.498** |

Table 2: Results unscrambling SIND visual stories[50, 2]. Captions are provided in the correct order; models must arrange the images temporally. MERLOT performs best on all metrics by reasoning over the entire story, instead of independently matching images with captions.

**3**. **Temporal Reordering**. We have the model order the image frames in a video, forcing it to explicitly learn temporal reasoning and giving it an *interface* to measure such temporal reasoning. Here, 40% of the time, we randomly pick an integer $i$ between 2 and $N$ (the number of segments provided to the joint encoder). Then we randomly scramble $i$ video frames chosen at random, by replacing the segment-level position embeddings (e.g. `[image_t]`) for that frame with a random and unique position embedding, e.g. `[image_unk_0]`). These random position embeddings are learned, and separate from the 'unshuffled' position embeddings. This allows the model to order each 'shuffled' frame conditioned on frames provided in the correct order (if any).

To compute the reordering loss, we extract hidden states from each frame at the `CLS` token position. For each pair of frames, we concatenate their hidden states $h_{t_i}$ and $h_{t_j}$ and pass the result through a two-layer MLP, predicting if $t_i < t_j$ or $t_i > t_j$. We optimize this using a cross-entropy loss.

### 3.4 Pretraining MERLOT

We pretrain our model for 40 epochs over our video dataset. We preprocess the dataset into examples with sequences of $N$=16 video segments each, each containing up to $L$=32 BPE tokens.[3] The language-only encoder computes contrastive representations given this entire sequence, its total length is thus 512 tokens. To save memory, we provide the joint vision-language encoder 4 groups of $N = 4$ segments each. At an image training resolution of $192 \times 352$, the joint model's sequence length is 396 tokens. To combine the losses, we multiply the contrastive loss by a coefficient of $0.25$, which we found scaled its gradient magnitudes to roughly the same magnitude as the Mask LM loss.

We train the model using a v3-1024 TPU pod, at a batch size of 1024 sequences (or 16k segments) in total. This pretraining process on this hardware takes 30 hours. We provide additional information about hyperparameters and experimental setup in Appendix E.1.

## 4 Experiments: Transferring MERLOT to Downstream Tasks

In this section, we explore MERLOT on 14 different tasks, covering vision-language reasoning on static images as well as videos; we present analysis and ablations to dig deeper into our performance.

### 4.1 Image tasks

**VCR.** We consider VCR [123], a task and dataset where models must answer commonsense visual questions about images. These questions, about e.g. 'what might happen next' or 'what are people's intentions,' force MERLOT to transfer video-level understanding to the world of single images.

VCR provides additional 'referring expression' information to models in the form of bounding boxes around named entities. For example, if `Person1` is referenced in the question, the location of `Person1` is also given in the image. We provide this information to models by drawing (in pixel space) a

---

[3]To train the model on as much data as possible, we merged together the segments of short videos, and split up longer videos, such that all preprocessed examples in our dataset have exactly $N$=16 video segments.

| Tasks | Split | Vid. Length | ActBERT [127] | ClipBERT$_{8x2}$ [67] | SOTA | MERLOT |
|---|---|---|---|---|---|---|
| MSRVTT-QA | Test | Short | - | 37.4 | 41.5 [118] | **43.1** |
| MSR-VTT-MC | Test | Short | 88.2 | - | 88.2 [127] | **90.9** |
| TGIF-Action | Test | Short | - | 82.8 | 82.8 [67] | **94.0** |
| TGIF-Transition | Test | Short | - | 87.8 | 87.8 [67] | **96.2** |
| TGIF-Frame QA | Test | Short | - | 60.3 | 60.3 [67] | **69.5** |
| LSMDC-FiB QA | Test | Short | 48.6 | - | 48.6 [127] | **52.9** |
| LSMDC-MC | Test | Short | - | - | 73.5 [121] | **81.7** |
| ActivityNetQA | Test | Long | - | - | 38.9 [118] | **41.4** |
| Drama-QA | Val | Long | - | - | 81.0 [56] | **81.4** |
| TVQA | Test | Long | - | - | 76.2 [56] | **78.7** |
| TVQA+ | Test | Long | - | - | 76.2 [56] | **80.9** |
| VLEP | Test | Long | - | - | 67.5 [66] | **68.4** |

Table 3: Comparison with state-of-the-art methods on video reasoning tasks. MERLOT outperforms state of the art methods in **12** downstream tasks that involve short and long videos.

colored highlight around the referenced entity (Appendix E.3.1), this differs from prior works (that integrate these entities into detection architectures).

Our results on the three VCR settings, in comparison to other models at the same ('base') scale, are given in Table 1. Our model outperforms these other models, that all learn from exclusively static images (paired with captions and supervised object detections).

**Unsupervised ordering of Visual Stories.** To probe our model's ability to do out-of-the-box commonsense reasoning over events in images, we next consider the Visual Storytelling dataset [50, 74]. Each story in this dataset contains five images and captions in a certain order; the order tells a joint narrative between the captions and the images. Past work has considered unshuffling image-caption pairs [2], but we take a slightly different approach in this work to avoid language-only biases, which can rely on discursive clues to order text [27, 102]. In our formulation, models are given the captions in sorted order, and must match frames to the captions. Our formulation disarms language-only baselines, while still allowing us to quantify MERLOT's capacity for commonsense temporal reasoning.

We compare MERLOT with two strong out-of-the-box baselines for text-image matching: CLIP [89], which encodes each caption and image separately and computes similarity through a dot product, and UNITER [22] which jointly represents each image/caption pair, and is trained in part using a 'text-image matching' objective. We use our temporal reordering loss to find the most probable ordering of the video frames (Appendix E.1.1); for CLIP and UNITER we compute a maximum-weight bipartite matching [63] over the pairwise image-text similarity scores.

Results over 5K stories are given in Table 2. MERLOT's performance in comparison to the algorithms trained from image-literal caption pairs suggests that, with no fine-tuning, our model has strong capability to reason about past and future events expressed in collections of temporal visual stories.

## 4.2 Video Reasoning

We report results on 12 video reasoning tasks: TVQA [64], TVQA(+) [65], VLEP [66], MSRVTT-QA [117], MSRVTT-Multichoice [121], LSMDC-Multichoice, LSMDC fill-in-the-blank QA [110, 92], ActivityNetQA [122, 45], TGIFQA [52], and DramaQA [23]. We apply MERLOT to these tasks in the same way. We sample a sequence of 5 to 7 still frames from each video clip, initialize new parameters only to map the model's pooled `CLS` hidden state into the output labels, and finetune MERLOT with a softmax cross entropy loss; see Appendix E.2 for details.

As shown in Table 3, for all these datasets MERLOT sets a new state-of-the-art. Given the diversity of tasks and the strengths of the comparison models, these results provide strong evidence that MERLOT learned strong multimodal and temporal representations.

## 4.3 Ablations

We present ablations over VCR and TVQA+ to study the effect of several modeling decisions.

| Training setup | VCR | TVQA+ |
|---|---|---|
| One segment ($N$=1) | 73.8 | 75.2 |
| One segment, attention masking | 73.5 | 74.5 |
| Four segments | 74.1 | 73.3 |
| 🍷 Four segments, attention masking | **75.2** | **75.8** |

(a) **Context helps together with attention masking.** Pretraining on more segments at once improves performance, but more context can encourage language-only representation learning. Attention masking counteracts this, giving an additional 1 point boost.

| Training setup | VCR | TVQA+ |
|---|---|---|
| No contrastive V-L loss | 57.5 | 67.6 |
| No temporal ordering loss | **75.5** | 75.6 |
| 🍷 All losses | 75.2 | **75.8** |

(b) **Contrastive V+L loss is crucial.** Removing it makes performance drop significantly; the temporal ordering loss is not as important for downstream finetuning.

| | VCR |
|---|---|
| No boxes | 74.8 |
| 🍷 Drawn-on boxes | **79.4** |

(c) **Drawing on bounding boxes helps**, suggesting that our model uses it to decode the 'referring expression' information (e.g. `person1`).

| Dataset | VCR |
|---|---|
| Conceptual ∪ COCO | 58.9 |
| HowTo100M | 66.3 |
| 🍷 YT-Temporal-180M | **75.2** |
| HowTo100M-sized YT-Temporal-180M | 72.8 |
| YTT180M, raw ASR | 72.8 |

(d) **Diverse (video) data is important.** Applying our architecture to caption data leads to poor results. Our model performs better on HowTo100M, yet still below our (more diverse) YT-Temporal-180M, even when controlled for size. Using raw ASR (vs. denoised ASR) reduces performance.

| # epochs | VCR |
|---|---|
| 5 epochs | 75.2 |
| 10 epochs | 75.9 |
| 20 epochs | 77.0 |
| 30 epochs | 78.5 |
| 🍷 40 epochs | **79.4** |

(e) **Training for longer helps,** with performance increasing monotonically over training iterations.

Table 4: Ablation study on the validation set of VCR question answering ($Q \rightarrow A$) and TVQA+, in accuracy (%). We put a 🍷 next to the configurations we chose for MERLOT.

**Context size**. Table 4a shows the effect of varying the number of segments $N$ given to the joint vision-and-language encoder during pretraining. In the first two rows, we provide only a single video segment ($N$=1) to the model.[4] In this limited regime, we find that our 'attention masking' approach (preferential masking of tokens that were highly attended-to by the contrastive language-only encoder) does not outperform a strong baseline of masking spans randomly [54]. Yet, when we expand the sequence length to $N$=4 segments/128 tokens, our masking becomes more effective, improving by 1 point over the baseline. This supports our hypothesis (Section 3.3.**2.**) that text-only shortcuts become increasingly viable with length, and that our attention-masking approach counteracts them.[5]

**Losses.** In Table 4b, we ablate the losses. We find that the contrastive frame-transcript matching loss is crucial to performance, suggesting that an explicit objective is critical for the (randomly initialized) image backbone to learn visual representations. The temporal ordering loss appears less critical for downstream tasks; it helps for TVQA but performance drops slightly for VCR. Thus, we find that it helps primarily as an *interface* by which we can query the model about temporal events (i.e. for the story ordering experiments); the model might be learning this information from other objectives.

**Drawing bounding boxes.** Table 4c shows the effects of providing grounding information to VCR models by drawing boxes. Performance drops 5% when they are removed, suggesting that they help.

**Dataset source**. In Table 4d, we investigate pretraining MERLOT on two datasets beyond YT-Temporal-180M. First, we train on 3 million static image-caption pairs from Conceptual Captions [99] combined with MSCOCO [71]; for fair comparison, we train for the same number of steps as 5 epochs on our dataset. The resulting model achieves 58.9% accuracy on VCR. We suspect this might be due to 1) a smaller context window (Table 4a), and 2) overfitting (5 epochs on YT-Temporal-180M corresponds to 300 epochs on the caption data). Because our vision pipeline is trained from scratch, the scale of the curated/supervised image pairing corpora is a concern.

We next investigate the impact of video selection, comparing YT-Temporal-180M with HowTo100M [80]. To control for number of videos, we train for an equivalent amount of steps: 5 epochs on our dataset, 30 epochs on HowTo100M, and likewise 30 epochs on a 'HowTo100M-sized YT-Temporal-180M'. Using diverse YT-Temporal-180M data vs. only instructional videos improves VCR performance by 6.5 points. This suggests that the how-to domain is limited in terms of visual

---

[4]We keep the effective batch size the same, so that we use $4\times$ the number of sequences at $\frac{1}{4}$th the length.

[5]Additional qualitative analyses of the attention patterns produced by the language-only encoder are in Appendix C.1; we find that highly attended-to tokens are typically more 'visual', and, thus, masking them may make the Masked LM objective require more cross-modal reasoning.

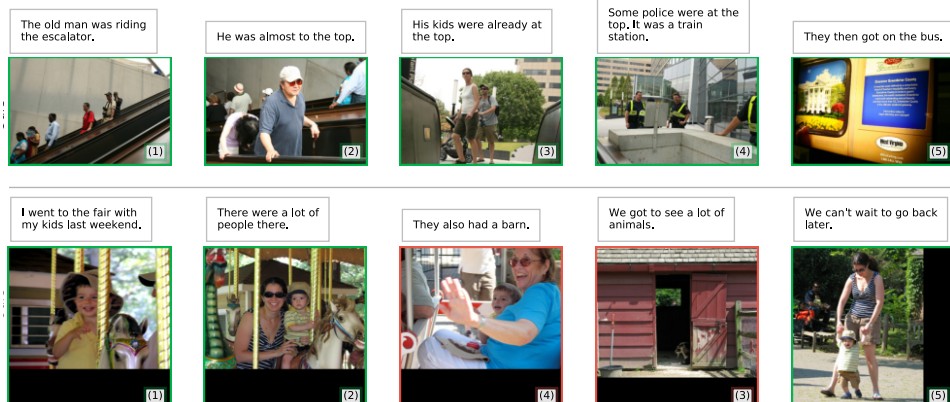

Figure 3: Zero-shot story ordering (same setup as Table 2). MERLOT performs temporal commonsense reasoning accross frames. In the first row, it uses 'the old man' mentioned to identify the 'kids' as parent-aged; in the second, it identifies riding a merry-go-round as an activity that takes a while.

phenomena covered, and that other domains (like web dramas and VLOGs) provide helpful signal for tasks like VCR [47]. Using all the data gives an additional 2.4-point performance boost.

Last, we investigate our choice to preprocess the YouTube ASR text with a language model (adding punctuation, etc); using 'raw ASR' instead of this preprocessing reduces performance by 2.4 points.

**Pretraining longer**. Last, in Table 4e, we investigate the effect of pretraining MERLOT for longer. The performance increases monotonically and doesn't begin to plateau, which suggests that had we pretrained MERLOT for *even* longer, its performance could improve even further.

### 4.4 Qualitative examples

In Figure 3, we show two qualitative examples of MERLOT's zero-shot story ordering capability. More examples (and a comparison with the best-scoring baseline, CLIP [89]) are in Appendix C.2. The examples here show that MERLOT has a strong understanding of events, transcending individual frames. In the first row, it orders the story correctly, performing vision-and-language coreference across several frames (e.g. frames and captions 2 and 3 use 'he' to refer to 'the old man' only mentioned in the first caption). Without resolving this coreference (establishing the subject as an elderly family member), it seems unlikely that anyone would describe the adults in frame (3) as 'kids.' Investigating the attention patterns of MERLOT (Appendix C.3) backs up this claim; they show that MERLOT frequently addresses video tasks by merging attention across (distant) video segments.

MERLOT gets the second row 'wrong', but for an interesting reason. It reverses the order of frames (3) and (4), which groups the merry-go-round pictures together – even though caption (3) mentions a barn. This seems to capture the temporal commonsense intuition that people might ride a merry-go-round for a while, i.e., it is not an atomic event [25].

## 5    Conclusion, Limitations, and Broader Impacts

We introduced **M**ultimodal **E**vent **R**epresentation **L**earning **O**ver **T**ime (MERLOT). We trained the model through a combination of self-supervised objectives on 6M YouTube videos, in service of learning powerful multimodal representations that go beyond single frames. The model achieves strong performance on tasks requiring event-level reasoning over videos and static images. We hope that MERLOT can inspire future work for learning vision+language representations in a more human-like fashion compared to learning from literal captions and their corresponding images.

There are several potential limitations of MERLOT that would make for promising avenues of future work, including: 1) exploring finer-grained temporal reasoning pretraining objectives vs. frame ordering e.g., a temporal frame *localization* within transcripts; and 2) learning multilingually from non-English videos and communities on YouTube.

Like other pretraining work, MERLOT risks some potential negative impacts. We discuss these in more detail below, in addition to the steps we took to reduce these harms.

## 5.1 Data collection and privacy.

As with other corpora gathered from the web used for pretraining data, YT-Temporal-180M contains publicly available content posted by users. We thus shaped our data gathering and release strategy to minimize inherent privacy and consent harms (Appendix A.5). Perhaps most importantly, we plan to only share video IDs for download, following a release strategy from prior work [1, 80] and giving users the right to opt out of not just YouTube, but our dataset as well.

## 5.2 Social biases.

The curation choices we made in this work could cause the model to exhibit undesirable social biases – *for this reason, along with others, we do not advocate for deployed use-cases.* For example, 30% of the data selected for by our filtering pipeline was local broadcast news (uploaded to YouTube). Including these news videos seems to perform better than filtering them out and only using how-to videos (Table 4b), however, there are risks when training on them. Local broadcast news (at least in the US) dedicates significant time to covering crime, sometimes in a racist and sensationalized manner [38, 29, 44]. Indeed, running a topic model over our data identifies several 'crime' categories (Appendix B). Past work has shown correlation between watching local news and having more explicit racialized beliefs about crime [28]; it seems likely therefore that training models on this data could teach them learn the same racist patterns.

Additionally, there are inherent social biases on YouTube – and treating these videos as equivalent to 'the world' [111] can embed hegemonic perspectives [42, 114, 13]. Most popular YouTubers are men [30] and video practices emerging on YouTube are often gendered [83]. YouTube also has problems with hate, including radical alt-right and 'alt-lite' content [90]. These problems – as with other problems in representation and power – are themselves amplified by the 'YouTube algorithm' [15] that recommends content to users. Though we downloaded videos independently of YouTube's recommender system, by filtering based on what content has views, we are implicitly filtering based on this algorithm. The dynamics of YouTube (i.e., which videos get popular/monetized) influence the style and content of videos that get made and uploaded to the platform; this in turn shapes and is shaped by culture more broadly [104].

## 5.3 Dual use.

The video QA tasks that we studied carry risk of dual use, through possible downstream applications like surveillance [91, 128]. It seems unlikely that purely technological fixes and defenses – which themselves can be problematic [40] – could resolve these dynamics. Studying how well video-level pretraining enables surveillance applications might be an important avenue for future work, if only to inform stakeholders and policymakers about these risks.

## 5.4 Energy consumption.

The pretraining that we used in this work was expensive upfront [105]. Our results suggest that scaling up the amount of data and compute that we used might yield additional performance gains – but at increased environmental cost. To pretrain more efficiently, we used a much more lightweight architecture (in terms of FLOPs) than is standard for today's vision and language models. We hope that our public release of the model (for research use) can further amortize this cost.

## 5.5 Synthesizing these risks.

With these issues in mind, we release MERLOT and YT-Temporal-180M for researchers. We view our work, and our research artifacts, to be part of a larger conversation on the limits of pretrained 'foundation models' [17]. These models have broad impact to real-world areas like healthcare, law, and education. At the same time, these models have significant risks, including the harms that we outlined. We believe that further academic research into this video-and-language pretraining paradigm is important – especially to probe its limits and possible harms. We hope that our paper, code, and data release can contribute to this direction.

## Acknowledgements and Funding Transparency Statement

We thank the anonymous reviewers for their helpful feedback that improved this work, along with Oren Etzioni and Gabriel Ilharco. Thanks also to Zak Stone and the Google Cloud TPU team for providing access to the TPU machines used for conducting experiments, and for help with the computing infrastructure. Last, but not least, thanks to all the YouTubers who share interesting videos with the world. This work was funded by DARPA MCS program through NIWC Pacific (N66001-19-2-4031), and the Allen Institute for AI.

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
