## Supplemental Material

We present the following items in the supplemental:

**a**. Data collection information (Section A)

**b**. An exploration of the data in our corpus (Section B)

**c**. Qualitative analysis of model representations (Section C)

**d**. An exploration of the intermediate visual representations (Section D)

**e**. Hyperparameters and experimental setup used for all experiments (Section E)

**f**. A Datasheet [37] for our YT-Temporal-180M dataset (Section F)

## A    Collecting Videos and Transcripts from YouTube

We adopt the following high-level process to collect YouTube videos and their accompanying transcripts:

**a**. Collect channel pages that are likely to cover visually-textually grounded events (A.1),

**b**. Download videos from each channel, while filtering out videos without English ASR captions, or unlikely to have (changing) real-world scenes and objects (A.2),

**c**. 'Denoise' the transcripts – using a language model to rewrite transcripts in a style more similar to written English, as opposed to spoken English (A.3),

**d**. Last, align words in the transcript to video frames, and extract the segments for pretraining (A.4).

As we will discuss in more detail in the following subsections, we designed our strategy to preserve user privacy as much as possible – an imperative when constructing a corpus on public-facing multimodal data. We conclude with a high-level summary of these privacy-preserving decisions, as well as about our release strategy (A.5).

### A.1    Collecting channel IDs + video IDs

The first stage in our pipeline was to collect YouTube video IDs that could potentially be relevant for learning visual-textual relationships. We opted to search for interesting *channels* rather than search for videos directly, as we found the API limits for searching for videos somewhat restrictive. Once a channel was downloaded, we could then download its videos.

We found channels using YouTube's auto-generated 'topic' pages, corresponding to entries in FreeBase like 'Science' or 'Home Improvement.' We identified 18 of these topics, and retrieved the IDs for all channels that were linked to by each topic page. We also used YouTube channels that appeared in the VLOG dataset [35], as well as a selection of viral 'How-To' and 'Cooking' channels. Last, we searched YouTube for concrete nouns, using the object list from MSCOCO ('baseball', 'snowboard', etc.) as a starting point; we retrieved channel IDs for each video that appeared.

Channels on YouTube often feature other (often similar) channels; so we downloaded more channel IDs by performing a graph breadth-first search over the initial set of channels. We identified 50k channels total and filtered out any more 'personal' channels (with fewer than 10k views between all videos). Last, we gathered all video IDs that came from our list of channels, which left us with 27 million video IDs, which formed our final candidate list.

*Privacy implications.* Our high-level goal was to preserve user privacy by mainly using popular (and more monetized) YouTube videos and channels in our dataset, as opposed to personal ones. The YouTube search algorithm helped us do that, by ordering results (in part) by the popularity of a video / channel. Downloading all videos from a channel, and filtering out channels with fewer than 10k views, favors popular content (like for celebrities, professional YouTubers, and cable news stations). Our analysis in Appendix B shows this strategy was largely successful.

*Connection with HowTo100M.* As discussed in the paper, we used both a diverse selection of YouTube videos (coming from this process), as well as the video list from HowTo100M [80]. We simply

concatenated the video IDs from HowTo100M with the video IDs from this searching step. This means first, that the HowTo100M videos were also filtered by the next steps (and thus our copy of HowTo100M is slightly smaller than the original), though we found that the filtering step had minimal impact on those videos (that were already filtered by [80]). Second, it means that the HowTo100M videos do contain some instructional videos from less-popular channels. Our intuition here is that this might be okay from a privacy standpoint: few of these people are discussing personal topics; a typical example might be a grainy video of somebody baking cookies. Nonetheless, given the scale that we operated at ourselves, we tried to be more cautious with the filtering.

## A.2 Filtering out videos

After retrieving a set of video IDs, our next step was to download ones likely to be appropriate for pre-training MERLOT. Not all videos would are likely to work well: many videos have no spoken words, are not in English, or otherwise do not have automatically-generated (ASR) captions. Likewise, many videos are not grounded: some just have still images (like podcasts), some are of people talking to each other or to the camera, and many are of people playing video games. Our intention was to filter out these videos, ideally without having to download them (so as to conserve bandwidth).

For each video ID, we perform the following steps:

- Downloading info: YouTube allows us to download the video metadata separately from each video. We do this first as the video info file is much smaller than the video itself. We thus first (try to) download this file. We exit here if one of the following conditions are met:
  - the video was removed,
  - the video is categorized as a 'Gaming' video,
  - the video does not contain any English ASR captions,
  - the video is over 20 minutes long (and thus might be overly expensive to download).

- Inspecting thumbnails: the YouTube API has a hidden feature that allows us to download four thumbnails [35]; in terms of bandwidth usage, this is often much cheaper than downloading the whole video. We use these thumbnails as a proxy as to whether the entire video is likely suitable for pretraining.[6] We trained a lightweight MobileNet-V2 CNN [93] to score whether a COCO object class is present in an image or not, using a sigmoid cross entropy loss. We exit here if one of the following conditions are met:
  - the CNN classifies fewer than four COCO objects as being 'present' over the four frames, using a minimum threshold of 30% probability for an object to be counted as being 'present.' This is mainly to recognize scenes with people, as opposed to animations, landscape footage, or blank/placeholder slides.
  - The average cosine similarity between all feature representations (computed by the classifier) is over 0.9; this allows us to skip videos that have no visual variance (like a person sitting in front of a camera for the whole video, or an album cover while a song is playing).

- Downloading the video: if we have not exited yet, we download the video.

## A.3 Denoising ASR Captions

One concern with pretraining on ASR is that written text may differ from spoken text: thus, when transferring to downstream tasks based on written corpora, models pretrained on spoken transcriptions may not transfer well. Also, ASR generated by YouTube does not include punctuation or capitalization. Furthermore, ASR transcripts can contain errors, e.g., by mistranscribing rare words/proper nouns and instead predicting incorrect, but similarly pronounced, words. And finally, YouTube's ASR system sometimes attempts to translate text from a different language to English, which is sometimes successful, but other times produces nonsense.

---

[6]Note that YouTube thumbnails are also (algorithmically) curated: when thumbnails aren't hand-selected by the uploader, YouTube's thumbnail selection algorithm selects high quality, clear frames. `https://ai.googleblog.com/2015/10/improving-youtube-video-thumbnails-with.html`

We aim to sidestep these issues by using a language model to 'denoise' ASR text, as well to filter out excessively noisy transcripts. We use a GROVER-Large language model to do this [124], as it was exclusively pretrained on written text from news articles. Then, we finetuned it in a sequence-to-sequence setting to 'denoise' ASR.

We created data for our 'denoising' task using the following procedure. Given an article from RealNews [124], we would trim it to 600 BPE tokens, and perform the following corruptions:

- We lowercase all text, and remove all punctuation.

- For each word (splitting by whitespace), we replace it with a random word 1% of the time. Within this 1%, 25% of the time, we use the CMU Pronouncing Dictionary[7] to swap-in a word with identical pronunciation (to simulate mistranscriptions), and 75% of the time we use a random sequence of BPE tokens of the same length as the actual word.

- For each word, 1% of the time we insert a 'filler word' before it, such as 'umm,' 'hmm,' or 'yeah.'

The model was trained to generate the 'noisy' news article, followed by a 'START' token, then the original 'clean' news article, and then an 'END' token; all using a standard cross-entropy loss. We prioritize learning the 'clean' text by multiplying the loss on the initial 'noisy' tokens by $0.01$. We trained this model using a batch size of 256 sequences of maximum sequence length 1536, a learning rate of 1e-5, and 80k steps.

The result is a model that not only attempts to fix mistranscriptions and corruptions, but also adds punctuation and capitalization. The model also produces an estimated likelihood of the ASR caption track, which we later use to filter out videos with very low quality ASR transcripts, e.g., poorly translated transcripts.

We apply the model to each video's transcript that survived the described filtration, breaking up long transcripts into groups of 512 tokens. These groups are handed as input to the model, and Nucleus Sampling (with $p$=0.9) [48] is used to generate a cleaned transcript for the group. We exit, filtering out the entire video, if any group has a perplexity of over 200. Finally, we concatenated all the groups together to form a 'clean' transcript.

## A.4   Putting everything together: aligning videos and cleaned transcripts to frames

To recap, at this stage in the pipeline, for each video, we have the video file, along with the original ASR transcript (with words, as well as timestamps for each word), and the cleaned ASR caption (without timing info). To estimate timing info for the clean transcript, we align the noisy and cleaned transcripts on a word-by-word level using Dynamic Time Warping [85]; word-word distance is computed using Levenstein distance. The timing estimate for a cleaned token was computed as the average of the noisy tokens assigned to it in this alignment.

Finally, given a video and its cleaned, per-word timed transcript, we sought to extract corresponding video frames – the data format we rely on for pretraining. We start with (empty) buffers of at most $L = 32$ tokens for both the original, and noisy transcripts. We loop through the (aligned) clean and noisy transcripts, and add the tokens to their respective buffers. If adding the next word would cause the buffer to exceed $L = 32$ tokens in length, we commit the segment – returning the noisy ASR text, along with the clean text, and timing information. We then extract a frame from the video corresponding to the middle of that segment. We do this until the end of the video. We use the GPT2 BPE encoder for this [97, 88], as was also widely adopted in later work (e.g. RoBERTa [72]).

Not all videos fit neatly into 16 segments, which was the format we used for training. Thus, we merged segments from videos shorter than 16 segments, and for longer videos, we split them into multiple examples. We didn't use any video sequence-level padding: all of our dataset examples have 16 valid frames, even though we did include padding at the token level (so many segments had fewer than $L = 32$ tokens).

---

[7]http://www.speech.cs.cmu.edu/cgi-bin/cmudict

### A.5 Summary - scraping while preserving privacy

As we discussed in the sections above, we tailored our scraping process to protect user privacy. It should be mentioned here that we focused on *public videos*. Possibly due to cues of engagement like view/subscriber counts, users on YouTube appear to understand the privacy implications of uploading a 'public' video [55], differentiating YouTube from more private venues, like email and social media. Under Marwick and boyd [78]'s framework of *networked privacy*, when web users (particularly those with less viewership) upload public videos, they are often '*being in public* without *being public*.' The idea behind this distinction is that web users, understanding that their content might be visible to others, tend to avoid sharing overly private data (like their phone number or date of birth); the information that they do share is often *encoded* (i.e., referring to a friend by their first name, not their full name). Finally, we took extra steps to filter out more 'personal' videos (without many views); our analysis in Appendix B shows this strategy was largely successful.

An additional aspect of our approach, as it relates to privacy, was our decision to use a diverse selection of channels. We did this to minimize risks of models 'overfitting' to specific individuals – a risk evidenced by a large GPT2 model memorizing users' phone numbers [18]. We believe that training a base-sized model in a large- and diverse-data regime minimizes many of the harms in this case; that said, the risk in the multimodal (video) space is unclear as of yet, and more research is needed.

Finally, we do not plan on releasing videos for download, only their IDs, following a strategy from prior work [1, 80]. This gives users an explicit 'right to be forgotten' not just from YouTube, but our data as well. We understand that this might make exact reproducibility difficult; we address this by releasing code for our filtering process. Thus, if in the future, if $N$ videos get deleted from YT-Temporal-180M, a practitioner can download $N$ new YouTube videos that pass through the same filters that we used.

## B   Data Exploration

Curating large pretraining corpora necessitates some ad-hoc decisions, e.g., what data to search for, what data to keep/discard, etc., and our work is no exception. The described data extraction pipeline contains several heuristics that we developed based on our subjective experiences (and per-step, heuristic validations) curating the corpus. While it isn't computationally feasible ablate each stage of this pipeline (and examine each decision's effect on downstream performance), we seek to quantify some basic of the properties of the corpus.

**Validity Check**   We randomly sampled 100 videos from the corpus, and answered the following basic questions for each of the videos: **Q1:** *Does the video contain language utterances?* **Q2:** *If so, is the language primarily English?* **Q3:** *Is the video an instructional video, i.e., is it an attempt to teach the viewer how to undertake a task?*[8] **Q4:** *What type of entity created the video: a small youtuber (<10K subscribers); a medium youtuber (<100K, >10K subscribers); or a large youtuber (>100K subscribers); a news station; or a media company.* **Q5:** *Is the video a music video?* **Q6:** *Is the video a video game commentary?*

Of the 100 examined videos, none were music videos or video game commentaries (Q5/Q6). The videos were mostly not instructional (84%) (Q3) and mostly in English (86%) (Q2); non-English videos nonetheless can have an English ASR track provided by the YouTube API if the spoken language is transcribed by YouTube via its auto-translate feature. And while all contained language utterances (Q1), at least one translated transcript had a very low quality transcription, which was only loosely semantically related to the underlying content. Finally, the most common video creators were news studios (29%; e.g., local news channels); big YouTubers (26%; e.g., popular vloggers), and media companies (24%; e.g., Major League Baseball). Also included, but in lesser proportion, were small YouTubers (8%), and TV studios (1%; e.g., official movie trailers).

**Content Exploration**   What topics are covered by the corpus? We randomly sampled 55K video transcripts, and ran an LDA topic model [16] implemented in MALLET [79] with 100 topics. We used a vocab size of 25K word types that appear in at least 25 transcripts, but in no more than 10% of

---

[8]A similar definition was proposed in [47].

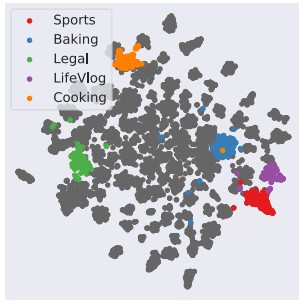

Figure 4: TSNE of topic distributions for 7K sampled documents.

| Sports | goal win match points ball games goals played players |
| Baking | sugar mix cup butter recipe flour oven dough bowl |
| Legal | court law justice judge investigation report prison |
| LifeVlog | excited vlog tomorrow literally camera bed yesterday |
| Cooking | sauce cook oil chicken salt garlic pepper cooking |

Table 5: Several common topics, derived from the transcripts of YT-Temporal-180M, represented by the most common words of those topics.

transcripts. The topics suggest diverse coverage, e.g., topics about specific sports (boxing, soccer), US and world politics, fashion, construction, fantasy settings, nail painting, etc. We use TSNE to visualize the per-document topic distributions, and color a sample of documents according to their top topic in Figure 4 (topic details in Table 5).

Overall, the topical coverage of YT-Temporal-180M, at least according to a topic model trained on the transcripts of a sample of videos, is broader than comparable-in-size video corpora like HowTo100M [80]. And, experiments in the main paper demonstrate that this diversity is apparently helpful for a number of downstream tasks.

## C   Qualitative Analysis of Model Representations

In this section, we provide more qualitative analysis about the representations learned by MERLOT.

### C.1   Analysis of the language-only encoder, and attention masking during pretraining

Early on in this project, when inspecting qualitative examples, we observed that using BERT-style masked language modeling [27] – choosing 15% randomly selected BPE tokens as the prediction targets, and replacing them with MASK 80% of the time, or a random token 10% of the time – produced overly easy examples.

This has been observed by other work in the text-only setting: when long words get partially masked, it is often easy to recover the missing BPE token from the context, which motivated Joshi et al. [54]'s choice to mask out entire spans instead. However, our goal in multimodal pretraining is different. We want the model to learn grounded representations of events, such that even when we scale up the number of segments given to the model, the model has to construct a multimodal representation of *what happened*. Thus, in our setup, we wanted to encourage masking out *highly visual* words, to learn cross-modal representations.

Instead of masking randomly, recall that we used the attention weights produced by the language-only encoder (trained to match a sequence of captions to individual frames) to inform which tokens to mask. While we do not claim that these attention weights provide a full explanation of the model behavior [51, 98], they do play some role in the model's decision [116], and we find that our masking strategy improves performance on downstream tasks by around 1% (Table 4), versus a SpanBERT baseline [54].

We show qualitative examples that seem to back up our hypothesis in Figures 5 and 6. In Figure 5, for instance, the video shows a VLOG of an adult playing with children and talking the camera. Tokens flagged by our approach as having high attention weights (being in the top 20% of all tokens in the sequence, in terms of other positions attending *to* that token) include concrete words like 'scissors' and 'toys.' Even though scissors are not shown in the selected frames, that word might be a good prediction target, insofar as it might complete a picture of what is going on in the first few frames: somehow, the adult is able to open the package with the child's toy, which could require scissors.

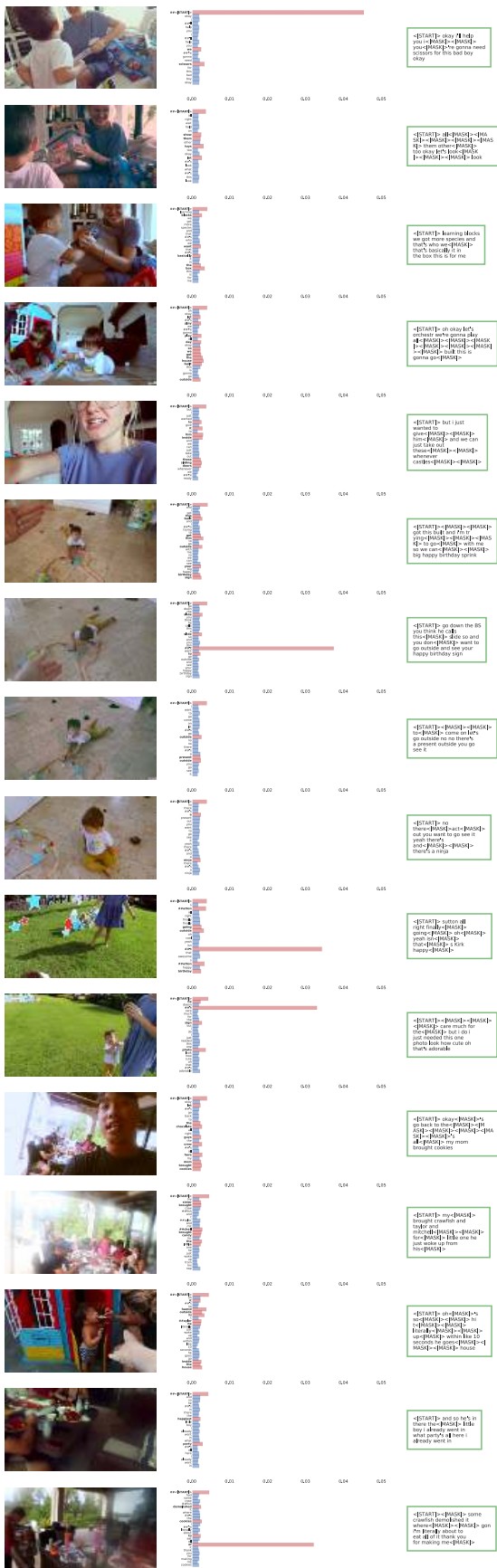

Figure 5: Attention masking for a video of 16 frames. Our model's image encoder learns image representations independently for each frame. A language-only encoder model takes in the entire transcript (with 32 words at most per frame) and computes hidden representations for each segment. The language encoder thus takes advantage of the inherent contextuality over time; each individual caption is not enough to understand the frame in isolation.

We use the language encoder's attention weights to mask out words. Tokens that are highly attended to (with the overall attention weights in the middle column) are shown in red and **bolded**. These tokens tend to be more grounded, e.g. the word 'toys' in the second row. The final input to the joint vision-and-language model is shown in the third column. We mask out highly attended-to words (except special tokens like 'START'), 50% of the time, which makes the pretraining objective much more visual than masking out random words (often fillers like 'on' or 'okay').

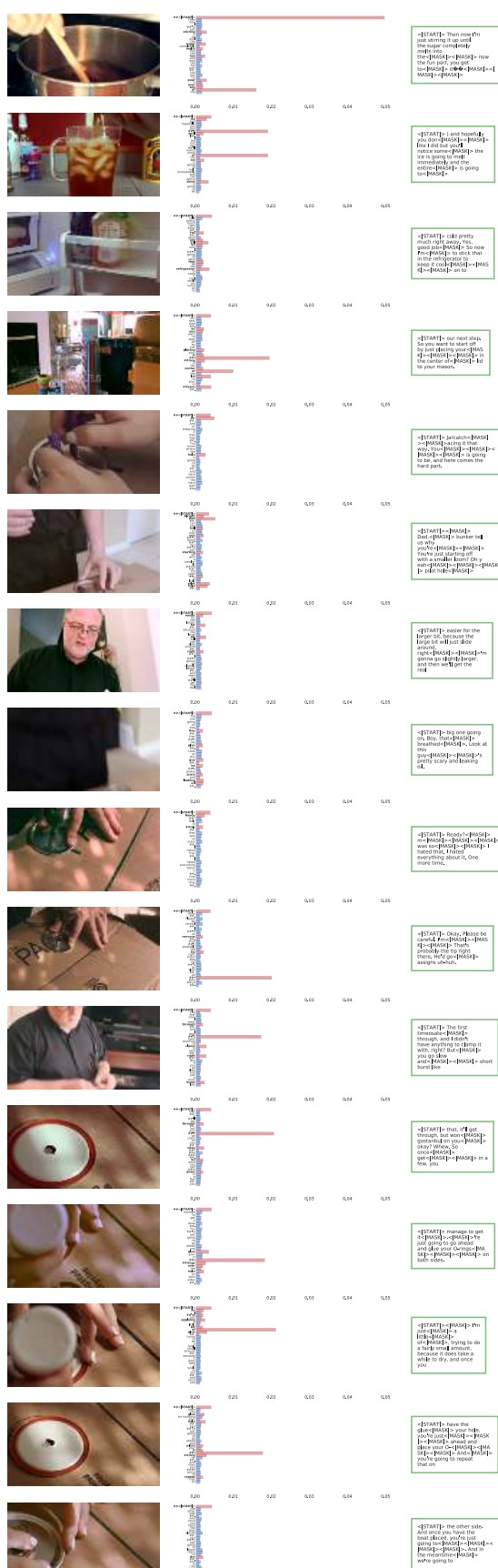

Figure 6: Another example of our masking approach, the same format as Figure 5. This shows an instructional video. Note the highly attended to tokens that get masked out (like 'ice', 'O-ring' and 'lid.') Seeing those objects in the image (not just through reading about them) is key to understand what the video is about – someone making iced tea in a mason jar.

|  | Constant | RSPNet [21] | $\mathcal{MERLOT}$-VizBranch | CLIP `ViT-B/16` [89] |
|---|---|---|---|---|
| UCF-101 [103] | 1.1 | 61.8 | 74.9 | 87.1 |
| HMDB-51 [61] | 2.0 | 42.8 | 49.6 | 62.4 |

Table 6: Linear probing classification accuracy of a $\mathcal{MERLOT}$'s intermediate visual representations (higher=better).

Additionally, in Figure 6, showing an instructional video for both making iced tea and putting it in a sealed mason jar, concrete nouns such as 'o-rings' get masked out.

Nevertheless, there are still several cases where the model seems to assign attention weights to apparently non-visual tokens. The model places a lot of attention on the START token, a pattern noticed by prior work as well [24], perhaps because we pool representations from those positions (for matching with the video frames). However, we never select the START token for masking in our work, so this might not highly affect the learning signal. Perhaps more strangely, language-only encoder seems to attend highly to the final token in contractions (like 't and 's). It is not clear to us whether these represent something important visually, or noise; we leave a more in-depth investigation of this phenomenon to future work.

### C.2 More qualitative examples for zero-shot story ordering

In this section, we show more examples of $\mathcal{MERLOT}$ unshuffling visual stories in SIND [50, 33]. We compare our model's zero-shot results (using the logits from its temporal-ordering objective) to CLIP's [89] independent matching of each caption with each image (using the Hungarian algorithm to find the best-scoring assignment [63]).

In Figures 7 and 8, we show expanded versions of Figure 3, comparing to CLIP. The examples show that $\mathcal{MERLOT}$ has a strong understanding of events that transcends individual frames. Unlike $\mathcal{MERLOT}$, CLIP can only match captions independently to images, so in the first row it struggles to connect 'his kids' with the middle-aged children of 'the old man' In the second row, it matches the barn image with the caption 'they also had a barn', while it is unable to keep all the merry-go-round images together (as $\mathcal{MERLOT}$ does).

We show additional examples in Figures 9 and 10. Our model provides a reasonable ordering to the 'kayaking' example (Figure 9), which is evident of multimodal script knowledge: first, people have to get ready to go kayaking (which they do on land!) and then they go out onto the water, and finally come back. The ordering of the tennis match (Figure **??**) seems reasonable as well. Unlike CLIP, $\mathcal{MERLOT}$ groups together frames (3) and (4) – the players first serving the tennis ball, and then awaiting the return.

### C.3 Attention patterns

Finally, we show examples of the attention patterns produced by $\mathcal{MERLOT}$, when it reasons over both vision-and-language content at a video level. Plots are shown in Figure 11. Overall, the model frequently links together visual regions with similar concepts in text, even when they get mentioned far away in time.

Though these attention patterns should be taken with a grain of salt, as they are not necessarily explanatory of the model's decision [51, 98], we find it promising that the model attends globally over all frames and captions – rather than ignoring one modality or ignoring the temporal dimension. We leave further investigation of the model's attention patterns and behavior to future work.

## D  Linear Probe of Intermediate Visual Representations

Our goal with $\mathcal{MERLOT}$ was to learn about situations expressed through videos and language. However, as it includes a vision encoder that we trained from scratch, a reasonable question is how this visual encoder compares to other encoders (e.g., that were trained through image captions). To this end, we performed linear probing experiments over two activity recognition datasets: HMDB-51

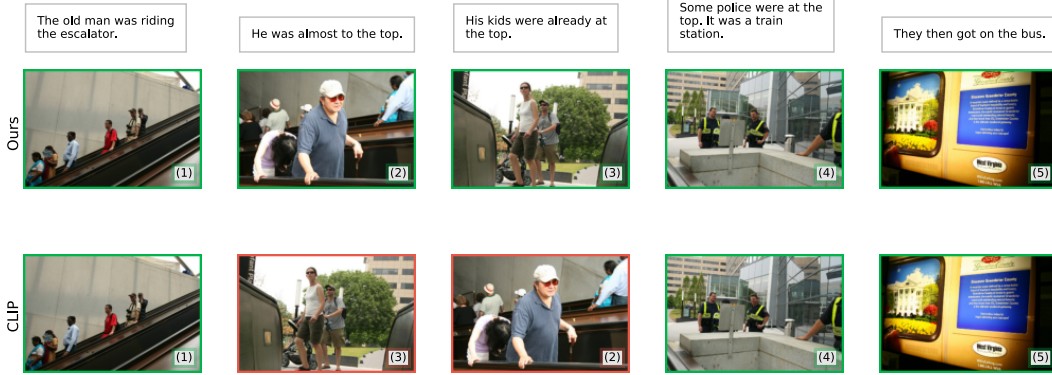

Figure 7: Zero-shot story unscrambling; continuation of Figure 3 with the CLIP baseline [89]. MERLOT successfully orders the story, performing cross-modal coreference over several images to note that 'He' in image (2) refers to 'the old man' mentioned in (1). The narrative that MERLOT generated also makes sense at an event level: people are riding the escalator, *then* they get to the top, *then* they exit and do something else; maximizing caption-image similarity of all pairs independently misses this event-level coherence.

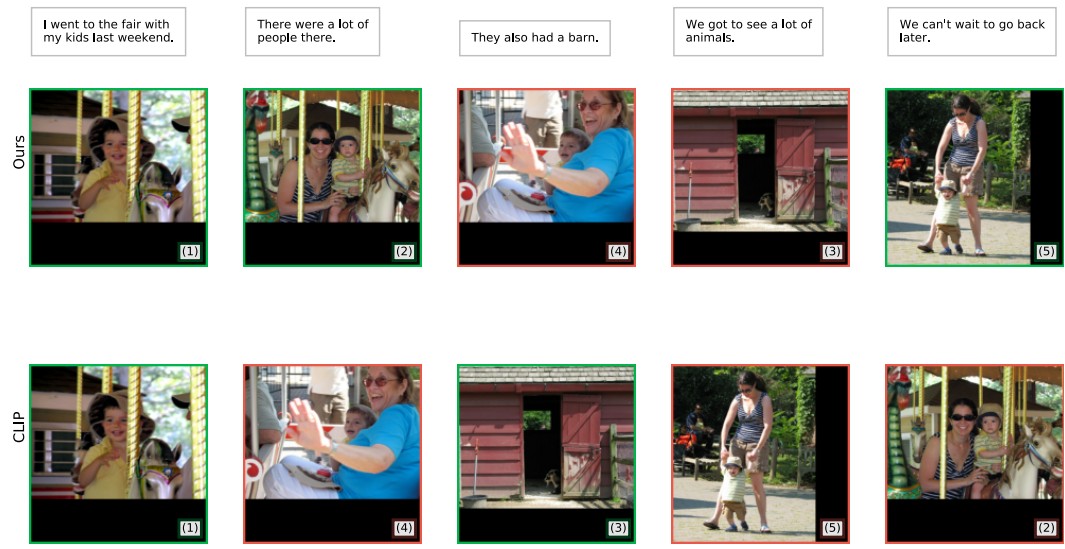

Figure 8: An incorrect story unshuffling example – but for an interesting reason. Frames (1), (2), and (4) all involve people riding a merry-go-round, and MERLOT keeps them together even though the ground truth story labels have the 'barn' image, (3), in between.

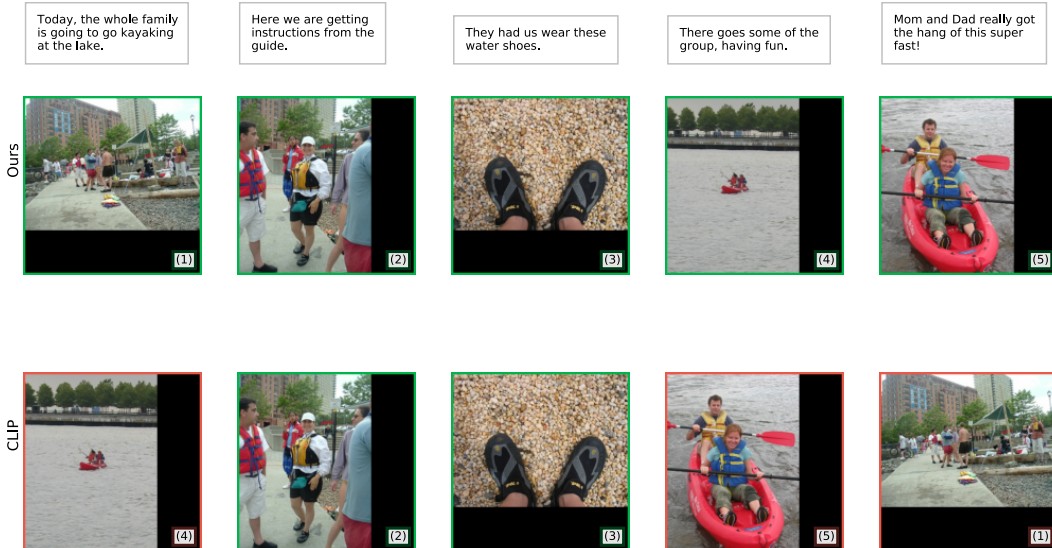

Figure 9: A second zero-shot story ordering example. MERLOT unshuffles the frames, while grouping together frames (1) and (2) – which make sense as they are in the stage of the event where they are *preparing to go*. CLIP instead puts frame (4) first, which matches caption (1) indepedently, but doesn't make sense temporally in context.

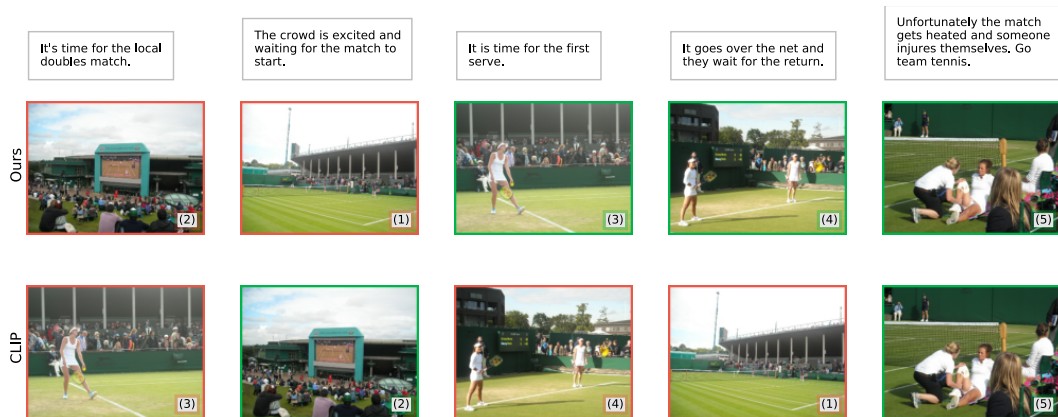

Figure 10: A second zero-shot story ordering example. There are a variety of potential 'reasonable' orderings for this example; both models get this one 'incorrect.' MERLOT's ordering suggests someone first looking into the tennis match on the outside, and then cutting to watch the match more closely. On the other hand, CLIP switches between a shot of someone serving, back to the outside TV, and then inside again.

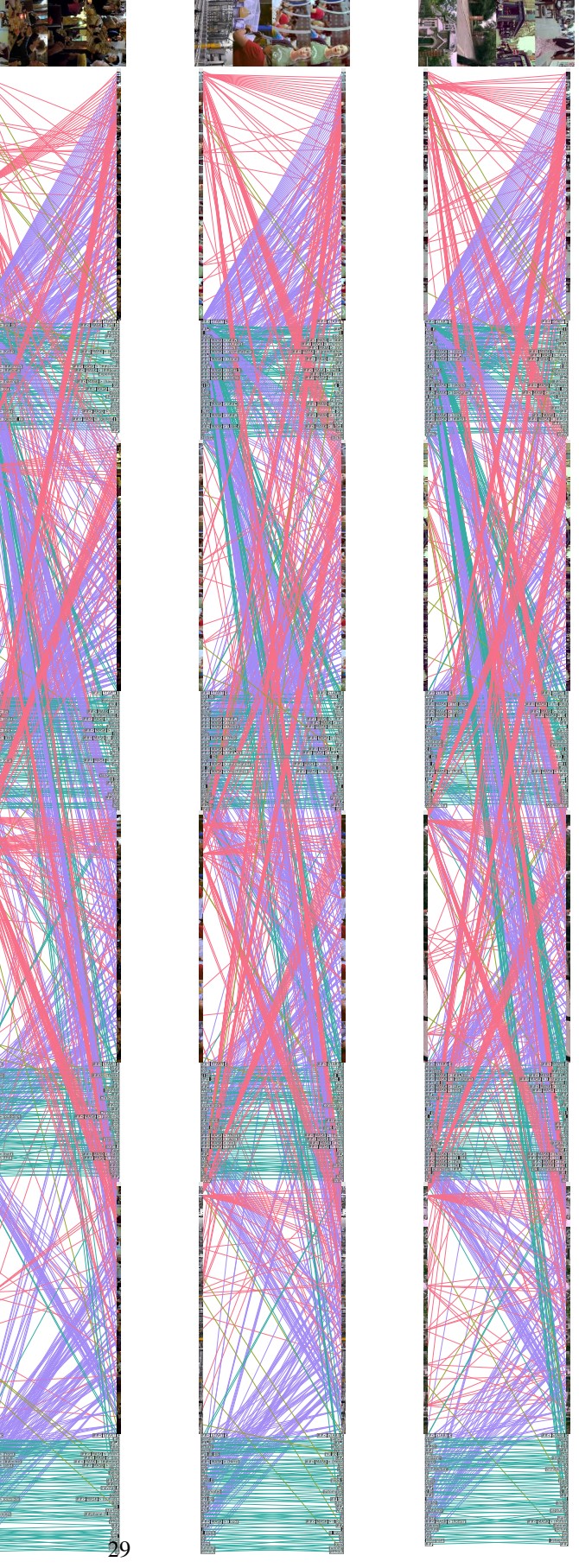

Figure 11: Additional qualitative examples of MERLOT's attention patterns, aggregated over all layers of the joint vision-language encoder. Cells on the top attend to cells on the bottom; we only show three attention edges per query, so as to reduce clutter.
In red, we show visual patches attending to other visual patches; in gold, we show tokens attending to tokens, and in purple we show patches attending to tokens.
The first row seems to show a tourist in the United Kingdom. In the third segment, the narrator discusses a 'gothic style house' even though only the gate is shown in the frame; those tokens attend highly to the house when it is shown in the fourth frame.
The second row shows someone at a factory for Dr. Bronner's Soap. The factory worker in the third frame seems highly attended to, particularly by the tokens 'applied by hand' which appear in the second caption.
The third row shows a dinner party. The first caption mentions 'nice food' but no food is shown in the first frame. Interestingly, the model has these tokens attend to the final frame, where food is shown.

[61] and UCF-101 [103]. These tasks are 51 and 101 class classification tasks, respectively: they challenge algorithms to predict which human activity is present in a video clip. Following prior work, for both datasets, we average over the three standard train/test splits. We evaluate in the linear probe setup, where models represent video clips as a single fixed vector, and a linear maximum entropy classifier is trained on top, freezing the rest of the model's parameters.

In addition to a random prediction baseline, we compare against [21]'s RSPNet reported results (they use a 3DResNet-18 backbone pretrained on Kinetics400), and CLIP `ViT-B/16` [89]. For $\mathbb{MERLOT}$ and CLIP, we extract a single central frame from each video, and extract a feature vector from it. For $\mathbb{MERLOT}$, we represent the frame as the concatenation of the two `[CLS]` tokens (one was for the image-transcript alignment task, the other was for passing to the joint encoder).

The results, shown in Table 6, show that CLIP performs best in this setup – though $\mathbb{MERLOT}$ does outperform an RSPNet baseline. At first, this might appear surprising, as $\mathbb{MERLOT}$ was trained on web videos, which might be closer to activity recognition datasets (as opposed to image captions). However, common benchmarks for activity recognition tend to have strong *object and background bias* – for example, to recognize the UCF action 'playing guitar,' it is sufficient to detect a guitar in an image (as guitars are unlikely to show up for the other activities like 'playing basketball') [70]. Temporal self-supervised learning from transcripts may not lead to as powerful zero-shot object detectors because speakers in videos may be less likely to state the obvious [41, 39], e.g., in this case, a speaker is probably unlikely to say 'I will now play a guitar while sitting in a chair.'

# E  Experimental setup and hyperparameters

## E.1  Hyperparameters used during pretraining

We used AdamW [73] with a learning rate of $3e - 4$, weight decay with value $0.1$, and set $\beta_2{=}0.98$. We used minimal data augmentation on the image frames. We randomly scale them between $1.125$ and $1.5$ times what would fit in our $192 \times 352$ resolution, and take a random crop. We use a random resize algorithm when doing this scaling, to make the model robust to different ways of preprocessing images [94]. Last, for 80% of images, we randomly jittered either their brightness or contrast to between $0.7$ and $1.3$ their original values, which we suspect did not play a major role in performance.

On the text side, we note that we have both the original copies of each transcript – what was retrieved from YouTube – and versions "cleaned up" by our denoisifier. We can use both kinds of transcript as additional data augmentation. However, although the words are time aligned, there might be inconsistencies if alternating between cleaned and noisy versions inside of a single video. Thus, for each iteration, we randomly choose either the 'clean' or 'noisy' ASR transcript and use that one.

To slightly speed up convergence, we initialize the joint vision-and-language model, and the word embeddings, with parameters from RoBERTa [72]. However, we suspect that due to the scale of our dataset and pretraining time, this might not have been required.

### E.1.1  Unsupervised Story Ordering

[20]

For the unsupervised scrambling of visual stories task, we did not do any finetuning on the SIND dataset [33, 50, 2]. However, there is a slight mismatch between the model that we pretrained initially, and the format of the task – the visual stories in the SIND dataset have 5 images and captions each, whereas we initially pretrained with at most 4 segments. We handled this discrepancy by pretraining $\mathbb{MERLOT}$ for 10 more epochs, using a peak learning rate of 2e-5, and a new resolution of 384 x 384. This slightly bigger size was to account for the (not necessarily) widescreen images in SortStory, as opposed to the (mostly) widescreen videos on YouTube.

Recall that $\mathbb{MERLOT}$'s pairwise loss is defined over pairs of segments. However, how to best combine these into a unified score for story ordering is an open question. To briefly explore this, during this additional pretraining of $\mathbb{MERLOT}$, we applied three variants of our temporal loss: one over caption-caption pairs, one over caption-frame pairs, and one over frame-frame pairs. We also experimented with randomly shuffling the captions as well, in the same way as the frames, we found however that this did not boost downstream task performance (perhaps because using shuffled captions as input incentivizes models to learn exclusively language-language interactions). The loss

is computed the exact same way everywhere; the only differences is that for caption-frame pairs, we have four options:

1. the caption (at $t_i$) and frame (at $t_j$) are of the same segment, so $t_i = t_j$,
2. the caption precedes the frame, so $t_i < t_j$,
3. the caption comes after the frame, so $t_i > t_j$,
4. the caption comes from a different video as the frame, so comparing $t_i$ and $t_j$ is undefined.

The model learns to distinguish between those four options with a cross-entropy loss. We found that using this version of the temporal loss over vision-language pairs produced slightly better results on story ordering (as judged on the validation set) compared with the loss applied over the frames. We hypothesize that this might be due to the additional '$t_i = t_j$' option allowing models to assign a probability to a frame-caption match, but are not sure. With this approach, to produce a unified score for (length-$N$) permutations $\sigma_L$ over the captions, and $\sigma_V$ over frames, we then sum over pairwise log-probabilities:

$$\text{score}(\sigma) = \sum_{i=1}^{N} \sum_{j=1}^{N} \log \begin{cases} p(\sigma_L(i) > \sigma_V(j)) & \text{if } \sigma_L(i) > \sigma_V(j) \\ p(\sigma_L(i) = \sigma_V(j)) & \text{if } \sigma_L(i) = \sigma_V(j) \\ p(\sigma_L(i) < \sigma_V(j)) & \text{if } \sigma_L(i) < \sigma_V(j) \end{cases}.$$

For story ordering, the order of the captions is always fixed: $\sigma_L = (1, 2, 3, 4, 5)$ and $N = 5$; we thus feed $\mathcal{MERLOT}$ captions with the correct order. However, the model should have no information about the order of the frames.[9] Recall that we handle this through position embeddings (3.3); e.g. one possible ordering might be

$$[\texttt{image\_unk\_3}], [\texttt{image\_unk\_2}], [\texttt{image\_unk\_4}], [\texttt{image\_unk\_1}], [\texttt{image\_unk\_5}],$$

and those position embeddings would get added to each frame, respectively. This allows the network to disambiguate between distinct frames even though no order is revealed. However, we found that the model was sometimes sensitive to the exact order of these position embedding tokens, and so for each example we randomly sampled two orderings and averaged the model's pairwise probabilities. We found no difference in performance when using more than two orderings. We hypothesize that this could be an issue with how (absolute) position embeddings are handled by Transformers, but are not fully confident; we leave a more thorough investigation for future work.

### E.2   Per-downstream fine-tuning details.

In this section, we discuss implementation details for finetuning $\mathcal{MERLOT}$ on downstream tasks. For each downstream task, given images $\boldsymbol{I}_{1:N}$ and language context $\boldsymbol{w}$, we first encode $\boldsymbol{I}_{1:N}$ via the image encoder. We concatenate this with word embeddings of $\boldsymbol{w}$, apply position embeddings, and feed the result into the joint vision-language encoder to extract joint representation. The input images $\boldsymbol{I}_{1:N}$ are either provided by the task or extracted from given video, where we uniformly select $N$ frames from the video clips (spaced evenly, so with an equal amount of time between sequential frames). For supervised tasks, we use as the 'head' a two-layer MLP from random initialization on top of the CLS token of the language context together with the rest of $\mathcal{MERLOT}$.

For downstream tasks, we note that we found it effective to finetune on different resolutions than what we used during pretraining. Our default image resolution here was $384\times 704$. To do this, we note that all parameters in the model remain the same, except for position embeddings on the image patches. We expanded the size of the position embedding matrix by initializing the upper-left-side 192x352 region from the pretrained model, and used random initialization for new position embeddings.

For all downstream tasks, we followed the standard training, validation, and test splits of the original datasets. We used the AdamW [73] optimizer, with $\beta_2 = 0.98$, and warmed up the learning rate linearly for the first 10% of iterations, followed by a linear decay of the learning rate (down to 0) for the remaining 90%. For regularization, we used L2 weight decay with a value of 0.01, and a dropout rate of 10%. For tuning other hyperparameters, we first did a larger random hyperparameter search over VCR, and used those hyperparameters as defaults for the other tasks. We used a batch size of

---

[9]Embarassingly, we found a slight leakage of this in the V1 of this arxiv paper which inflated the story ordering performance by a few percentage points (of pairwise accuracy), which we have corrected in this version.

64, and searched over learning rates in the range [1e-5, 2e-4] on VCR, we found that 1.2e-5 worked well, so we used it as the default for other tasks. We also trained with early stopping, validating every epoch and returning the best-performing model across epochs. Due to our choice of early stopping, we trained for a slightly larger-than-typical number of epochs (18 by default for every tasks, as we found training longer did not help on VCR).

We follow the standard evaluation metrics for these tasks, which is usually accuracy for QA-style configurations. Alongside brief descriptions of each downstream task, we provide hyperparameter and training details in the following section.

### E.3 Static Image Reasoning Tasks

#### E.3.1 VCR

VCR [123] contains two different subtasks: question answering (Q→A) and answer justification (QA→R), both of which are multiple choice questions over a given image. These subtasks are combined in the joint Q→AR metric, which requires a model to both pick the right answer and the right rationale for the model to get a question 'right.' VCR has 290k questions over 110k movie scenes.

As mentioned in the main text, VCR provides bounding boxes around entities, with explicit groundings between those entities and references in questions. We draw colored highlights around the referenced entity directly in the image, with consistent mapping between color code and entity name (e.g. person1 with red box, person2 with green box, etc). Though no text is written on the image, because we always associate each string (e.g. person1) with a deterministic color, the model can learn through finetuning to associate that color with the entity. Figure 12 illustrates one such example.

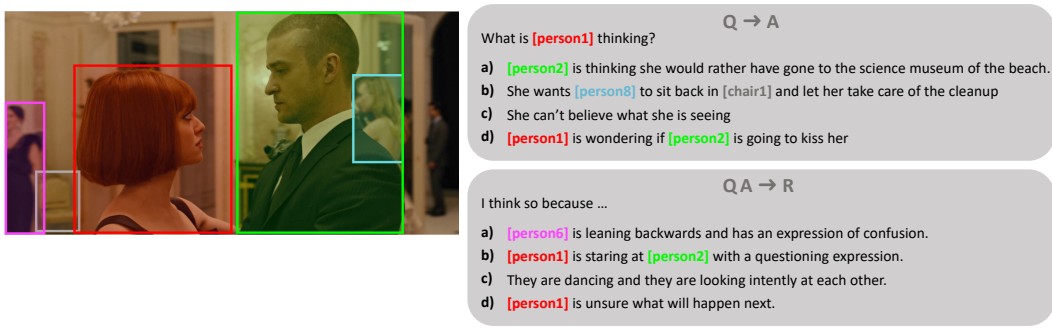

Figure 12: A VCR example with highlighted image. The image with the drawn-on boxes is what we pass to models.

We jointly finetune $\mathbb{MERLOT}$ on Q→A and QA→R, with two separate MLP heads. We concatenate the question (the question and the ground truth answer) and each answer (rationale) choice from the four possible answer (rationale) candidates. On-top of the CLS token of the question, we train the classifier to predict the confidence for each candidate to be correct with cross-entropy loss, and take softmax over four possible candidates for each question. We used a widescreen resolution of 384×704 set the batch size as 64, and train for 60k training steps, which is roughly 18 epochs. We started with this and then tuned the learning rate (from candidates chosen randomly); here, we found that a learning rate of 1.2e-5 worked well. We then used this learning rate as a default for the other tasks.

Note that our pretraining setup is different from other work. Previous works [22, 36, 119] conduct what they call 'second-stage pretraining' with VCR training data. Here, they use a masked language model objective over the VCR dataset (instead of answering the question correctly). In particular, UNITER [22] reports 2.8 % point performance boost due to the second-stage pretraining. We suspect that this might be because the caption data (that models like UNITER rely on) are quite different from VCR. We tried performing secondary pretraining and found it did not help. One possible reason might be that our large-scale pretraining corpus covers diverse and complex event space thus we don't need additional data domain adaptation.

|         | What (50K) | Who (20K) | How (2K) | When (677) | Where (250) | Overall |
|---------|-----------|-----------|----------|------------|-------------|---------|
| AMU [117] | 26.2 | 43.0 | 80.2 | 72.5 | 30.0 | 30.5 |
| VQA-T [118] | 35.5 | 51.1 | - | 81.0 | 43.5 | 41.5 |
| MERLOT | 37.0 | 52.9 | 85.3 | 79.2 | 42.8 | 43.0 |

Table 7: Per question-category results for MSRVTT-QA.

## E.4 Video Reasoning Tasks

**MSRVTT-QA** [117]

MSRVTT-QA is a question-answering task with 244K questions posed over 10K videos. For each video clip, we uniformly selected 5 image frames (spaced evenly through the video). We follow the protocols of the original work and use an answer vocabulary containing the most common 1K answers in the training set as answer candidates. The questions with out-of-vocabulary answer will automatically get wrong. We encode the answers in a one-hot fashion, and train 2-layer MLP classifier over all answer candidates with a binary cross-entropy loss on-top of the CLS token of the question. We train for 60k training steps with batch size 16. A few additional fine-tuning runs were conducted to examine the effect of changing the resolution from $384 \times 704$ to $704 \times 704$, a batch size of 16 vs. 32, and and using 1.5K answers instead of 1K, but none had much impact on validation accuracy. We undertook a light hyperparameter optimization over the validation set, wherein we considered 3 possible learning rates (**1.2e-5**, 6e-5, 2.4e-6), but the default worked best. MSRVTT-QA splits questions by type, and we report our per-type test set results in comparison to [117, 118] in Table 7.

**TVQA** [64]

TVQA is a multiple choice task with 152K questions posed over 21K video clips. For each clip, we uniformly select 6 image frames. We concatenate the question and each answer choice from the five possible answer candidates. On-top of the CLS token of the question, we train 2-layer MLP classifier to predict the confidence for each candidate to be correct with cross-entropy loss, and take softmax over five possible candidates for each question. We set the batch size as 64, and train for 35k training steps (roughly 18 epochs over the corpus). We used the default learning rate of 1.2e-5, and a resolution of $384 \times 704$.

**TVQA+** [65]

TVQA+ is a subset of TVQA, where bounding boxes are provided in video clips, linking depicted objects to visual concepts in questions and answers. TVQA+ contains 29.4K questions posed over 4.2K video clips. We uniformly select 6 image frames per video, and draw bounding boxes on each frame following the same manner with VCR. We train the classifier in the same way with TVQA. We trained with the same hyperparameters as TVQA, but for 16k steps (18 epochs still).

**VLEP** [66] VLEP is a binary choice task to infer which of the two events is more likely to happen next following the given video. VLEP contains 28.7K questions posed over 10K video clips. For each clip, we uniformly select 6 image frames. On-top of the CLS token of the event, we train 2-layer MLP classifier to predict the confidence for each event to happen next with cross-entropy loss, and take softmax over two possible events for each instance. We trained the model for 8k steps (18 epochs over the dataset), and with otherwise default hyperparameters.

**DramaQA** [23]

DramaQA is a multiple choice task with 17.9K questions posed over 23.9K video clips. For each clip, we uniformly select 6 image frames. We concatenate the question and each answer choice from the five possible answer candidates. On-top of the CLS token of the question, we train 2-layer MLP classifier to predict the confidence for each candidate to be correct with cross-entropy loss, and take softmax over five possible candidates for each question. We trained for 3.5k steps (18 epochs) with otherwise default hyperparameters. A few additional fine-tuning runs were conducted to examine the effect of changing the resolution between $384 \times 704$, $512 \times 512$ and $704 \times 704$, and we found $512 \times 512$ works the best for this task.

|  | Resolution | Batch Size | Max Epochs | Training Steps |
|---|---|---|---|---|
| VCR | 384x704 | 64 | 18 | 60k |
| MSRVTT-QA | 384x704 | 16 | 18 | 35k |
| TVQA | 384x704 | 64 | 18 | 35k |
| TVQA+ | 384x704 | 64 | 18 | 35k |
| VLEP | 384x704 | 64 | 18 | 18k |
| DramaQA | 512x512 | 64 | 18 | 18k |
| TGIF-Action | 384x704 | 16 | 56 | 70k |
| TGIF-Trans | 384x704 | 16 | 22 | 70k |
| TGIF-FrameQA | 384x704 | 16 | 56 | 70k |
| ActivityNetQA | 384x704 | 16 | 10 | 34k |
| LSMDC-FIB | 384x704 | 16 | 8 | 150k |
| LSMDC-MC | 384x704 | 16 | 12 | 80k |
| MSRVTT-MC | 384x704 | 16 | 12 | 80k |

| Common hyperparameters | |
|---|---|
| Learning rate | 1.2e-5 |
| Weight Decay | 0.01 |
| $\beta_2$ | 0.98 |
| Warmup ratio | 10% |

Table 8: Hyperparameters for finetuning on all downstream tasks. Common hyperparameters are shown to the left, and task-specific hyperparameters are to the right.

|  | Motion | Spatial | Temporal | Yes-No | Color | Object | Location | Number | Other | All |
|---|---|---|---|---|---|---|---|---|---|---|
| VQA-T [118] | 28.0 | 17.5 | 4.9 | 66.3 | 34.3 | 26.7 | 35.8 | 50.2 | 36.8 | 38.9 |
| MERLOT | 33.9 | 18.1 | 4.0 | 72.5 | 36.2 | 24.5 | 36.5 | 51.7 | 37.8 | 41.4 |

Table 9: Per question-category results for ActivityNetQA

## TGIF-QA [52]

TGIF-QA is web GIF VQA, which requires spatio-temporal reasoning from visual frames to answer questions correctly. We finetuned MERLOT on three tasks in TGIF-QA benchmark,

*Action* is defined as a multiple choice question about identifying an action that has been repeated in a video.

*Transition* is asking about transitions of certain states. The benchmark provides a multiple choice question about identifying the state before or after another state.

*FrameQA* is asking open-ended questions about the given video. The model selects answer from a dictionary of words, given a question in a complete sentence.

For each video clip, we uniformly select 5 image frames. We serialized 5 candidate answers and a question, where we put a special token QSEP between the candidate answers and question to concatenate them into one question. On-top of the CLS token of the question, we trained 2-layer MLP to predict the confidence of the five candidates with cross-entropy loss. We set the batch size as 16, and train for 70k training steps (*Action* : 56 epoch, *Transition* : 22 epoch, *FrameQA* : 28 epoch) for each task with 1.2e-5 learning rate. We used a longer training duration for each task as we found that performance increased when we did so (and we used the same number of training steps for each TGIF-QA task). All other hyperparameters were default.

## ActivityNetQA [45, 122]

ActivityNetQA [122] is a question-answering with 58K questions posed over 5.8K videos. For each video clip, we uniformly select 5 image frames. We use an answer vocabulary containing the most common 1K answers in the training set as answer candidates. The questions with out-of-vocabulary answer will automatically get wrong. We encode the answers in a one-hot fashion, and train 2-layer MLP classifier over all answer candidates with a binary cross-entropy loss on-top of the CLS token of the question. We set the batch size as 16, and train for 34K training steps for each task. We undertook a light hyperparameter optimization over the validation set, wherein we considered 3 possible learning rates (**1.2e-5**, 6e-5, 2.4e-6), but the default worked best. A few additional fine-tuning runs were conducted to examine the effect of changing the resolution from 384×704 to 704×704, a batch size of 16 vs. 32, and using 1.5K answers instead of 1K, but none had much impact on validation accuracy. ActivityNetQA splits questions by type, and we report our per-type test set results in comparison to [118] in Table 9.

## LSMDC FiTB QA [76, 92]

The Fill-in-the-blank (FiTB) task is, given a video clip and a sentence with a blank in it, to predict a single correct word for the blank. The test set includes 30,000 examples from 10,000 clips (i.e. 3 blanks for each description). For each clip, we uniformly select 5 image frames. We constructed answer vocabulary containing the most common word for blank in the training set as answer candidates. We replace the blank in the sentence with BLANK token, so the question query should be a blanked sentence with the special token. On-top of the CLS token of the blanked sentence query, we trained 2-layer MLP classifier to predict the word for the blank over answer vocabulary. We set the batch size as 16, and train for 150k training steps (8 epoch) with 1.2e-5 learning rate.

### LSMDC Multichoice [110]

Given a video query and 5 candidate captions, the task is to find the one that fits the query out of 5 possible candidates. The correct answer is the ground-truth (GT) caption, and four other negatives are chosen from other captions that have different activity-phrase labels from the correct answer. We randomly created 100,000 video and candidates pairs for training. For each video clip, we uniformly select 5 image frames. We put a special token QSEP between the candidate captions to concatenate 5 candidates into one question. At the end of the 5 captions, we put CLS token as an end of the question. On-top of the CLS token, we trained 2-layer MLP to predict the confidence of the five candidates with cross-entropy loss. We set the batch size as 16, and train for 80k training steps (12 epoch) with 1.2e-5 learning rate.

### MSRVTT Multichoice [121]

The task objective for the MSRVTT Multichoice benchmark is identical to those of corresponding tasks in the LSMDC benchmark [110]. The benchmark has 2,990 questions in total for the multiple choice test, using all the test video clips of MSR-VTT. For each test video. We finetuned our model on MSR-VTT train split, and evaluated on the evaluation set. We trained the same model specification as the LSMDC Multichoice task. For training, we set the batch size as 16, and train for 80k training steps (12 epoch) with 1.2e-5 learning rate.

## F  Datasheet for YT-Temporal-180M

In this section, we present a DataSheet [37, 12] for YT-Temporal-180M, synthesizing many of the other analyses we performed in this paper.

1. Motivation For Datasheet Creation
   - **Why was the dataset created?** In order to investigate learning events from videos – involving a collection of frames and captions over time, that together form a view about the world.
   - **Has the dataset been used already?** No.
   - **What (other) tasks could the dataset be used for?** Possibly other types of representation learning, with or without ASR captions.
   - **Who funded dataset creation?** This work was funded by DARPA MCS program through NIWC Pacific (N66001-19-2-4031), and the Allen Institute for AI.

2. Data composition
   - **What are the instances?** The instances that we consider in this work are videos, paired with ASR transcripts aligned over time.
   - **How many instances are there?** We include 6 million videos. The total length of all the ASR transcripts is 5 billion BPE tokens. Altogether, we extracted 180 million image frames from this data.
   - **What data does each instance consist of?** The instances have 'raw' video frames and text, which we preprocess through BPE tokenization and extracting frames for every 32 BPE tokens.
   - **Is there a label or target associated with each instance?** We only use the ASR captions as labels in this work, though it might be also possible to use auxiliary information (like tags or video titles).
   - **Is any information missing from individual instances?** No.

- **Are relationships between individual instances made explicit?** Not applicable – we do not study relations between different videos (e.g. made by the same creator), though this is a possibility for future work
- **Does the dataset contain all possible instances or is it a sample?** Just a sample.
- **Are there recommended data splits (e.g., training, development/validation, testing)?** We do not provide recommended data splits at this time, as this data was built only for pretraining rather than evaluation. We suspect that the data is large enough that overfitting is not a major concern.
- **Are there any errors, sources of noise, or redundancies in the dataset? If so, please provide a description.** Yes. YouTube ASR is often noisy, and though we presented a pipeline to correct some of these errors, there are many that we cannot fix.
- **Is the dataset self-contained, or does it link to or otherwise rely on external resources (e.g., websites, tweets, other datasets)?** The dataset is self-contained. However, we plan to only release the video URLs, rather than the videos themselves, so as to protect user privacy (allowing users to delete videos).

3. Collection Process

- **What mechanisms or procedures were used to collect the data?** We used the YouTube API and the `youtube-dl` library.
- **How was the data associated with each instance acquired? Was the data directly observable (e.g., raw text, movie ratings), reported by subjects (e.g., survey responses), or indirectly inferred/derived from other data?** The data was directly observable (from YouTube).
- **If the dataset is a sample from a larger set, what was the sampling strategy (e.g., deterministic, probabilistic with specific sampling probabilities)?** We used a probabilistic strategy with many heuristics, more details in Appendix A.
- **Who was involved in the data collection process (e.g., students, crowdworkers, contractors) and how were they compensated (e.g., how much were crowdworkers paid)?** Data collection was primarily done by the first authors of this paper.
- **Over what timeframe was the data collected? Does this timeframe match the creation timeframe of the data associated with the instances (e.g., recent crawl of old news articles)? If not, please describe the timeframe in which the data associated with the instances was created.** The data was collected from November 2020 to April 2021, even though the YouTube videos are often much older (dating back to when the platform was first created).

4. Data Preprocessing

- **Was any preprocessing/cleaning/labeling of the data done (e.g., discretization or bucketing, tokenization, part-of-speech tagging, SIFT feature extraction, removal of instances, processing of missing values)?** Yes, we discuss this in Appendix A: of note, we use a sequence-to-sequence model to 'denoise' ASR transcripts (Appendix A.3), BPE-tokenize text, turn everything into segments, and extract the middle image frame for each video segment.
- **Was the "raw" data saved in addition to the preprocessed/cleaned/labeled data (e.g., to support unanticipated future uses)? If so, please provide a link or other access point to the 'raw' data.** The raw data was saved, but at this time we do not plan to release it directly due to copyright and privacy concerns.
- **Is the software used to preprocess/clean/label the instances available? If so, please provide a link or other access point.** We will make our code public to support future research.
- **Does this dataset collection/processing procedure achieve the motivation for creating the dataset stated in the first section of this datasheet? If not, what are the limitations?** We believe our dataset does allow for study of our goal – indeed, it covers grounded temporal situations from a variety of domains – but with significant limitations. Some of the key ones we are aware of involve various biases on YouTube, which we discuss in Section 5.

5. Dataset Distribution

- **How will the dataset be distributed?** At this time, we plan to distribute all the metadata (transcripts, etc) that we used, as well as links to the YouTube videos that we used. We will do this on our website.
- **When will the dataset be released/first distributed? What license (if any) is it distributed under?** We will release it as soon as possible, using a permissible license for research-based use.
- **Are there any copyrights on the data?** We believe our use is 'fair use,' however, due to an abundance of caution, we will not be releasing any of the videos themselves.
- **Are there any fees or access restrictions?** No.

6. Dataset Maintenance

- **Who is supporting/hosting/maintaining the dataset?** The first authors of this work.
- **Will the dataset be updated? If so, how often and by whom?** We do not plan to update it at this time.
- **Is there a repository to link to any/all papers/systems that use this dataset?** Not right now, but we encourage anyone who uses the dataset to cite our paper so it can be easily found.
- **If others want to extend/augment/build on this dataset, is there a mechanism for them to do so?** Not at this time.

7. Legal and Ethical Considerations

- **Were any ethical review processes conducted (e.g., by an institutional review board)?** No official processes were done, as our research is not on human subjects, but we had significant internal deliberation when choosing the scraping strategy.
- **Does the dataset contain data that might be considered confidential?** No, we only use public videos.
- **Does the dataset contain data that, if viewed directly, might be offensive, insulting, threatening, or might otherwise cause anxiety? If so, please describe why** Yes – many of these videos exist on YouTube; we discuss this more in Section 5.
- **Does the dataset relate to people?** Yes.
- **Does the dataset identify any subpopulations (e.g., by age, gender)?** Not explicitly (e.g. through labels)
- **Is it possible to identify individuals (i.e., one or more natural persons), either directly or indirectly (i.e., in combination with other data) from the dataset?** Yes, our data includes celebrities, or other YouTube-famous people. All of the videos that we use are of publicly available data, following the Terms of Service that users agreed to when uploading to YouTube.