# OpenReview forum: "MERLOT: Multimodal Neural Script Knowledge Models"
_NeurIPS.cc/2021/Conference — NeurIPS 2021 Oral_

### Official Review · Reviewer_eAHy · 2021-07-15

**Rating:** 7
**Confidence:** 3

**Summary:**

In this paper, the authors propose a video-text based self-supervised learning objective to learn strong representations of temporal commonsense. They curate a large scale dataset of videos with transcribed captions (from ASR) from diverse everyday situations to learn stronger and diverse knowledge representations. They perform extensive experiments  on 12 downstream tasks of video question-answering and on the visual commonsense reasoning tasks and outperform state-of-the-art models. They also perform extensive ablation studies to show the importance of pre-training on video datasets compared to image based pre-training.



**Limitations And Societal Impact:**

The authors have discussed the limitations and impact of their work. One thing to consider is the cost while training on large scale video datasets versus on image datasets and how potentially that can be curbed by not hurting performance and understanding of the visual world.

**Main Review:**

The main contributions of the paper are an end to end vision and language model to learn powerful multimodal representations from videos and transcripts using no labeled data (no bounding boxes and or fine grained captions as in previous work) and a diverse large scale dataset of YouTube videos. They use three objectives to train the joint vision-language model - a contrastive frame caption matching, masked language modeling and temporal ordering of frames. The paper proposes a novel idea and is well written and shows extensive experiments.

1. The contrastive frame-caption matching loss considers the CLS token to compute the loss. How would the model perform by taking the average embeddings of the image matches instead of the CLS level tokens ? How important are the CLS level tokens for the image encoders ?

2. As both the contrastive and temporal re-ordering loss are over the image CLS token embeddings, is this supervision enough to learn good and meaningful image patch representations ([1,1] … [H,W]) ? What if we only have the image level embeddings from the image encoder?

3. Does the video-pre training work better just because the dataset is much larger than  the image ones?  Is it still possible to achieve good performance (reasoning and understanding) with large scale image datasets with strong cross modal attention and architectures ?



##Post Rebuttal

I have carefully read the authors' response and they address my concerns. I maintain my original rating.


**Time Spent Reviewing:**

3.5

---

> ### Author Response · Authors · 2021-08-10
> **Author response to eAHy**
>
> Thanks for your thoughtful review!
>
> ### CLS token vs Averaging + patch embeddings for the image encoder
>
> Thanks for the interesting idea! We didn't thoroughly investigate different pooling strategies. If accepted, as discussed in the response to aTvC (R3), we plan to include results on UCF101 and HMDB51 to investigate the effectiveness of our image encoder. We will try both pooling strategies in those experiments. And, we can investigate the semantics of the patches in this process, either using an interpretability method, or qualitatively, depending on the outcome.
>
> For the temporal ordering loss, this is an interesting idea also! However, we note that the temporal loss is applied over the outputs of the multimodal transformer; e.g. the model looks at the shuffled image frames in the context of the words that are said, to determine the right position. Thus, it might not directly translate to lower-level image/patch representations, but we are excited to explore this!
>
> ### Could we try training MERLOT on a very large-scale image dataset?
>
> We appreciate the suggestion, and would love to do this if such a large-scale image dataset was publicly released!
>
> Unfortunately, to the best of our knowledge, today the largest publicly available image-text dataset is ConceptualCaptions, which contains on the order of millions of image/text pairs. We trained on this and have results in Table 3b; an analogous ‘small’ video dataset might be HowTo100M, of which we used 800k videos; training on videos gives us 28 points of improvement on TVQA+ even in a ‘small’ setting.
>
> Larger datasets like ImageWebext (CLIP, OpenAI, 400M image/caption pairs) exist, but they are not publicly available. Still, we compared CLIP to MERLOT on the zero-shot VIST frame ordering task, and MERLOT does better (Table 2), even though our model only sees 180M image frames, versus CLIP’s 400M.
>
> These results together suggest the benefit of training on data with temporal dynamics (YouTube) vs. ImageWebText or ConceptualCaptions.
>
> ### Cost of training on video datasets versus image datasets
>
> Thanks for the suggestion, we’ll incorporate this into our limitations and impact section in revision.

---

### Official Review · Reviewer_aTvC · 2021-07-16

**Rating:** 8
**Confidence:** 5

**Summary:**

This paper introduces a new approach to learn from large collections of uncurated video and text obtained from Automatic Speech Recognition Transcript. The authors introduce a dataset of 6M YouTube videos that lead to better results than existing large scale video and language datasets. The proposed architecture consists of separate backbones adapted to images (ResNet50) and text (Transformer) followed by a joint Vision and Language transformer. The model is trained with three different losses:
* (i) a contrastive text-vision matching loss applied before the joint transformer,
* (ii) a masked language modeling loss (with a modified masking mechanism to bias towards visual words),
* (iii) a temporal ordering loss to learn to reorder vidoe frames.

**Ethical Concerns:**

No ethical concerns for that paper.

**Limitations And Societal Impact:**

Limitations and societal impact were properly addressed.

**Main Review:**

## Strengths

- Clearly written

- A new dataset that can be impactful for the community (important for the authors to confirm that the dataset will be released at least in the form of YouTube ids + narrations).

- State-of-the-art results on 12 different benchmarks

- Interesting innovations on the model side: biased masking strategy, contrastive loss coupled with Joint Transformer based approach.

## Weaknesses + clarification points

- More ablation studies: importance of the different losses is not properly ablated such as the temporal ordering loss does not seem to be ablated in the main paper. Similarly the novel masking technique is not compared to the default random masking strategy. Please include these ablations in the rebuttal.

- Confirm that the training dataset will be made available.

- I found the data collection description quite interesting in the Appendix. It might be worth surfacing this more in the main paper. In particular how important was the procedure to clean the transcripts? Have you ablated that?

- It would be nice to also evaluate the learned visual representations since they are trained from scratch. For example it would be nice to compare to [6, 56] on benchmarks such as UCF101 or HMDB51.

- Related to the previous point, would it be possible to perform zero-shot text-to-video retrieval on benchmarks such as YouCook2 or MSRVTT to compare to previous work? In particular most benchmarks are evaluated via specific finetuning while it would be Interesting to be able to evaluate the model without any adaptation.

- The model only takes as input single frame per segment, how can it then model dynamic scenes? Can the authors comment on that limitation and how to address it in future work?

- Minor comments and questions:

  * Typos: atttention -> attention (Table 3, (a)).
  * Sentence boundaries: how are the text boundaries exacty obtained? In L138 BPE is mentionned with 32 tokens but it seems that a sentence separator is also used. Are the two things combined somehow?


## Overall assessment

Overall, MERLOT is a good paper that introduces a good large scale pretraining datasets, interesting architecture innovations to deal with the scale of the data and obtains strong results on several benchmarks. As of now, I lean towards acceptance but request the authors to address the points mentionned above.

## Post rebuttal

After reading other reviews and rebuttals, all my questions have been addressed. I believe MERLOT is a strong paper that will likely be influential in the vision and language community.

Of particular interest I believe that showing that leveraging YouTube data can improve **image** tasks such as VCR is quite interesting. Also after the additional ablation requests, I believe this will help make the contributions in terms of architecture and losses clearer in the paper. I hope the authors will include this nicely in the final version.

**Time Spent Reviewing:**

4

---

> ### Author Response · Authors · 2021-08-10
> **Author response to Reviewer aTvC**
>
> Thanks for your thoughtful review!
>
> ### More ablation studies
> Thanks for the comment, we will include these ablations in revision and have attached them in the author response:
>
> * Transcript de-noising: Please see the response to CWuA for the ablation results. The denoising helps by over 2% on VCR.
> * Temporal objective: Please see the response to w8dn. We found the temporal objective to be most critical as an interface for the model’s ‘zero-shot’ commonsense script knowledge. It seems to slightly help on TVQA, however, it is possible that what it encourages the model to learn might also be learned implicitly by the other objectives.
> * Masking strategy: we have this ablation in Table 3a (“Four segments” is our method, but without using attention weights to mask). Our masking strategy gives a 1% boost in performance on VCR, and 2% on TVQA+.
>
> ### Dataset release
>
> We confirm that the training dataset will be made available. We want to do so in the most ethical way possible, so see our response to w8dn regarding this point as well.
>
> ### Comparison of image representations on visual-only datasets UCF101 or HMDB51 (instead of just vision+language benchmarks)
>
> Thanks for the suggestion! While not central to our focus on vision+language tasks, exploring action classification on the visual-side would offer valuable perspective on our trained-from-scratch image encoder. If accepted, we will include performance comparisons on these corpora in a camera ready.
>
> We will also investigate performance on zero-shot retrieval (like YouCook2 and MSRVTT). We note that our model was not designed for this task (as much of the parameter budget is for encoding video frames jointly with text), so this kind of adaptation might be nontrivial and models with stronger independence assumptions (like CLIP) might do better; we will investigate this further and discuss the results in the camera ready.
>
> ### How does our model handle dynamic scenes?
>
> Thanks for the comment -- we will discuss it further in the ‘limitations’ section of the paper. However, we would argue that in the context of learning multimodal script knowledge, giving the model a single frame might actually be a strength. The reason being is that our goal is for MERLOT to infer what’s going on in the world, temporally, through (partial) observations of both vision and language. Just like humans, a model of visual commonsense should be able to infer “what might happen next” from even a single image. If we gave the model every single frame, the model might not need to make these inferences about the ‘between’ frames, and so it might do worse on datasets like zero-shot SortStory.
>
> This is a strong hypothetical however, because passing vision models sequences of frames is expensive, especially at our scale (a finding argued also by ClipBERT). We think one exciting way to address this might be to have a more efficient (joint) encoding of dynamic scenes, and we will discuss this possible line of future work in revision.
>
> ### Minor comments (re: text-boundaries)
>
> Text boundaries: we indeed use BPE with 32 tokens. As described in a bit more detail in appendix A, we iterate through the video transcript and greedily fill up a segment with words until adding a new one would hit 32 tokens in length. The special "START" token for the segment, which occurs at the beginning, is added afterwards. We'll clarify this in revision.

---

> > ### Comment · Reviewer_aTvC · 2021-08-31
> > **Thanks for the answer**
> >
> > Just a message to ack that I read this response and that I am satisfied with the provided answers.

---

### Official Review · Reviewer_w8dn · 2021-07-18

**Rating:** 7
**Confidence:** 3

**Summary:**

The submission proposes a model and dataset for learning representations of multimodal events by training over YouTube videos with ASR captions. The dataset (YY-Temporal-180M) consists of 180 video segments pairs with words extracted from ASR.  The dataset is taken from 6 million YouTube videos and spans multiple domains and topics.  The model (MERLOT) is a transformer based model that is trained on image and language to 1) predict frame-caption match 2) recover masked words (masked language modeling) and 3) predict the temporal ordering of frames (given two frames f1, f2, does f1 come before or after f2).  For 1) a contrastive loss is used and the features for the caption for a segment is extracted not just from the segment but from the entire transcript (it is a contextualized representation for the segment). For 2), attention weights from a language-only transformer is used to favor masking of highly attended-to-tokens so to reduce masking of fillers such as 'umm' or 'yeah' that are common in spoken speech (SpanBERT masking is also applied for robustly against noise).
Experiments on 14 downstream tasks (mostly video QA tasks) demonstrates the proposed model/dataset/training objective combination outperforms existing state-of-the-art results on these tasks.   There are also ablations comparing the number of epochs to train, the dataset used for training, and training setup (different amount of context, different loss terms).

Main contributions
- MERLOT: a Transformer-based
- YY-Temporal-180M - dataset of frames and transcription from a filtered set of YouTube videos
- Experiments and ablations showing the combination of the proposed model and dataset can outperform prior models on 14 downstream tasks


**Ethical Concerns:**

The dataset contains videos involving people and potentially identifiable information.  There are concerns of privacy.  While the videos are uploaded to YouTube and public, it is unclear whether the individuals would be okay with videos of themselves and corresponding transcripts being used for research.

The authors documented the concerns and their plan for how to release the data in the appendix.

**Ethics Review Area:**

["Privacy and Security (e.g., consent)"]

**Limitations And Societal Impact:**

Limitations and societal impact is discussed.

**Main Review:**

Originality: Currently, there is many different models using transformer-based models for vision and language.  Despite some novel elements, the overall originality of this work is limited.

Quality: The works looks to be well executed with reasonable rationales for the design choices, and a good set of experiments.

Clarity: Overall, the paper is well written and well organized.

Significance: The dataset (if it can be publicly released) will be useful to foster additional research in this area.  The proposed model and experiments are also useful.

Strengths:
- The model is well-designed and experiments are well-executed
- The dataset (YY-Temporal-180M) would be useful for the community

Weaknesses:
- Poor description of how the proposed model (MERLOT) relates to prior work on pretraining with video and language.  It's unclear specifically where the novelty of the model lies.
- Lack of comparison with prior models (ClipBERT, ActBERT) pretrainined on the same data.  It is unclear whether pretrained on the same dataset (YY-Temporal-180M), the proposed model performs better at the specified tasks than models from prior work.
- The ablation studies is not that thorough.  For instance, while there is ablations on the contrastive VL loss and the attention masking, there is no ablation on the temporal ordering loss.
- The dataset does not seem like it will be easy to download as only links to the associated YouTube videos will be provided.  This also means that the videos may disappear over time.

Other comments:

Some of the claims in the introduction about learning commonsense and script-knowledge representation is not really substantiated - the main evidence for this is that the model outperforms other models on two very specific tasks (the VCR and SortStory tasks).

References should be proofread and updated so that the publication venue is clear and consistent.
- [20],[21] is duplicated
- [80]: "Leon Sigal" => "Leonid Sigal"

**Needs Ethics Review:**

Yes

**Time Spent Reviewing:**

2

---

> ### Author Response · Authors · 2021-08-10
> **Author response to Reviewer w8dn**
>
> Thanks for the feedback!
>
> ### Originality of our model architecture + comparison with CLIPBert and ActBERT?
>
> Indeed -- we designed our architecture around being deliberately similar to past works (a convnet, followed by a transformer), so we agree regarding architecture :)
>
> Our main modeling contribution in this work was not architecture, but rather, proposing a new paradigm “to learn powerful multimodal representations from videos and transcripts using no labeled data” (eAHy). This is where the key modeling differences are: unlike CLIPBert and ActBERT for instance, we don’t use a vision backbone trained on supervised data (ResNet on Visual Genome for CLIPBert, and a Faster-RCNN trained on Visual Genome for ActBERT).
>
> We agree that it is important, however, to compare against standard VisualBERT models like these -- and we believe that our current experiments allow three distinct lines of comparison:
>
> 1. Comparing against a baseline “images + captions” VisualBERT model, without object detection, and pretrained on our data. Pretraining such a model could be seen as a special case of MERLOT, though with only one frame (and associated ASR caption) being given to the model at a time. We have this ablation in Table 3a:
>     * If no contrastive V+L loss is applied, the baseline does poorly (57.5% on VCR Q->A).
>     * If our contrastive V+L is applied, even in a setting where models see only one frame+caption at a time, the result is 73.8% on VCR. We believe this is a novel result in the context of vision-and-language models; the earlier models do not run into this issue as they use supervised vision backbones.
>     * Our proposed method of giving the model 4 frames and captions at a time, gets 75.2% on VCR, suggesting that our pretraining objectives let it outperform prior work.
> 2. Comparing against a baseline “images + captions” VisualBERT model, pretrained on Conceptual Captions/COCO. We have this in Table 3b; training on a large collection of videos enables over 16 points of performance gain on VCR, which is one of the main messages of this work.
> 3. Comparing against past SOTA on benchmarks: We compare against ActBERT and ClipBERT on many of the same benchmarks that ActBERT and ClipBERT reported on (e.g. MSR-VTT-MC; TGIF-QA), and achieve new state-of-the-art results.
>
> We’ll revise the paper to better synthesize these directions, all of which suggest strong benefits of our video-based pretraining approach.
>
> ### Ablation of the temporal loss on downstream tasks (like VCR and TVQA)?
>
> Thanks for the suggestion! At a high level, we included our temporal ordering objective primarily as an “interface” to enable MERLOT to achieve strong results on multi-frame reasoning (on VIST). By training the model to score the “right” ordering of video frames, we can visualize how well it can reconstruct what might have gone on in the world -- a key aspect of how we as humans reason with multimodal script knowledge. We suspect that much of the learning during pretraining happens through the other objectives (e.g. predicting missing words requires some degree of understanding ‘what’s going on’ or ‘what might happen in between frames’). We will discuss this further in submission.
>
> Still, this raises an important question -- to what extent does temporal ordering allow MERLOT to do better on single-image tasks (like VCR) and video tasks (like TVQA)? We performed this ablation and report results (after 8 epochs, same as the paper) on these datasets’ validation sets:
>
> | *Model*                                |        *VCR Q->A*        |    *TVQA+*       |
> |----------------------------------------|--------------------------|----------------|
> |MERLOT (with temporal loss, 8 epoch)    |             75.1%        |      75.7%     |
> |MERLOT (without temporal loss, 8 epoch) |           75.5%          |      75.6%     |
>
> The results show slight improvement on TVQA, and a slight decrease on VCR, but both are likely within reasonable error bounds.
>
> We will report these results in the paper, along with an important caveat. We suspect that a key confounding factor might be the (shorter) duration of training used for the ablations (8 epochs) in comparison to the full model (40 epochs). In other words, the temporal loss randomly replaces the position embeddings of certain frames. After a long time training, the model is able to correctly learn this (e.g. after 40 epochs, the flagship model in the paper gets 79.6% accuracy on VCR validation and 81.6% on TVQA), but 8 epochs might be too soon. We will investigate this further and discuss these results in revision.
>
>
> ### What’s the best (and most ethical) way to release our dataset?
>
> We thank the reviewer for bringing attention to an important issue (that we have also spent a lot of time thinking about)! As mentioned by the other reviewers (w8dn; aTvC), there is likely to be significant interest in YT-Temporal-180M, so it is important to release it ethically.
>
> With respect to releasing data, there exists a tension between reproducibility (which would favor a full dataset release) and privacy (which would favor no release). We are eager to discuss the appropriate course of action with the reviewers and the program committee, including the reviewer from the ethics committee, if assigned.
>
> In any case, we took proactive steps to scrape data in a way that respects user privacy. We tailored our scraping process around downloading videos from large channels (e.g. professional-quality videos, manually-written captions); these videos tend to feature YouTube celebrities and thus using them in our academic data can be considered as ‘fair use.’ We double checked the effectiveness of our data curation strategy in Appendix B: over 92% of our videos come from media companies, or bigger-name youtubers with at least 10K subscribers.
>
> As an additional step to protect privacy, when releasing data, we seek to give users the ‘right to be forgotten’ i.e., the right to delete their data. Thus, we favor a release strategy that is similar to what has been used for prior datasets, like for YouTube8M (Abu-El-Haija et al 2016) and HowTo100M (Miech et al 2019). These datasets release YouTube IDs rather than the raw video files, which means that if a video is deleted by the uploader or removed by YouTube, the video will simply be unavailable when the API is queried. We plan to also release denoised video transcripts with the YouTube IDs, with anonymized channel IDs. This means that if the video is deleted from YouTube however, it will be difficult to link the transcript with the YouTube(r)’s identity.
>
> While we believe that ethical concerns ought to take priority, reproducibility matters as well. We suspect that future work might want to train on some variant of YT-Temporal-180M, as well as further investigate our data-curation decisions. We plan to address this by publicly releasing code for our filtering process (e.g. our MobileNet V2-based image filter, and our LM Denoiser), along with the queries we used. This allows for reproducibility while preserving the ‘right to be forgotten’ -- if in the future N videos get deleted from YT-Temporal-180M, N new videos can be used that pass through the same strict filtering process as those in the original data.
>
> Last, we agree that there are ethical concerns of using/releasing web data where users did not explicitly consent to their data being aggregated into training sets: like ImageNet, HowTo100M, and the CommonCrawl: YT-Temporal-180M aggregates without explicit permission; and like AlexNet, BERT, and CLIP, MERLOT's weights compress information from their underlying corpora. We are eager to further discuss the role of web-scraped data in ML pretraining, and how our work fits into that discussion.
>
> ### Other comments
> Thanks for these, we will fix them in revision (and clarify the connection between multimodal commonsense knowledge, and the datasets that we have today to evaluate this like VCR, SortStory, etc.).

---

### Official Review · Reviewer_CWuA · 2021-07-18

**Rating:** 7
**Confidence:** 4

**Summary:**

This paper makes several contributions: a large scale video-language dataset obtained from 6M Youtube videos; a end-to-end VL model that learns multimodal representations, MERLOT; and large number of experiments including ablations, zero-shot transfer, and fine-tuning to achieve a new state of the art on a host of popular video language tasks, as well as challenging image-text tasks such as visual commonsense reasoning.


**Limitations And Societal Impact:**

Paper discusses considerations of data collection and resulting social biases well in Section 6. Additionally, the Appendix includes a detailed "Datasheet" for the YT-Temporal-180M dataset.

**Main Review:**

*Strengths:*
1. A large new dataset to rival HowTo100M, YT-Temporal-180M, that is not limited primarily to instructional videos. I found the performance improvements somewhat surprising, considering the prevalent understanding that a generic Youtube video (non-instruction, non-vlog) need not have narration describing visual aspects shown on screen.
2. Another pleasant surprise is that the model achieves the performance by simply using a mid-frame of a video segment, instead of needing to rely on complicated 3D CNNs.
3. Thorough pre-processing and three important pretraining objectives probably contribute to much of the success. Unfortunately, it's not very clearly explained how much is gained from each of the three pretraining tasks (Weaknesses 1a).
4. Thorough experiments on a variety of tasks on video/vision-language understanding, achieving new SOTA.

*Weaknesses:*
1. While this is a strong submission, it would be useful to include a few additional ablations / experiments:
a) Impact of overall pre-processing steps vs. directly using all videos and the three pretraining tasks could be demonstrated perhaps on VCR and TVQA+ as done in Table 3. Seems to me that these are critical aspects of the contribution, and while I understand that they are costly experiments in terms of compute, are necessary to shed light onto what made the model actually work. Some combinations are already presented in Table 3a.
b) If similar dataset pre-processing steps are applied to HowTo100M, would the performance reported in Table 3b improve further?
c) The large jump in performance on VIST is probably owing to the "temporal reordering" task. Might be interesting to analyze performance without this objective (as indicated in ablation above).

2. Dataset:
a) The dataset filtering (Appendix A.2) seemed like it primarily focuses on videos of people showing a variety of common objects (filtered based on presence of at least 4 objects in COCO). This may filter out most documentary style videos from Youtube - but it doesn't affect performance since most of the downstream tasks are about people as well.
b) Some statistics such as number of seconds per segment, the distribution of videos by Youtube categories, etc. might be nice to include in the appendix.

3. VCR experiments. L249-251, while it's very interesting that the model can see the colored bboxes in the image and reason about them, it might be interesting to also try a baseline model (any of the "visual BERT" models) with such a strategy to properly benchmark improvement.

4. Minor:
a) L106 says "large body of work" on action forecasting, but only cites one.
b) L124, I think there might be space to mention the MobileNet-V2.
c) Are segments of the video that do not have a speech transcript ignored?
d) I'm not convinced by the word "knowledge" in the title. It gives the impression of a potentially symbolic component or at least an evaluation on such related knowledge graphs - Comet, Atomic, etc.


Note: Many experiments asked in this review will probably take time longer than that provided for the rebuttal. It's sufficient to include a brief plan instead.


### Post-rebuttal

Thanks to the authors for their response. I found that it explains several questions about the implementation details satisfactorily. I also briefly read all other reviews and responses and we all seem to agree that this is a good paper to accept.

I probably mis-framed the question about temporal ordering and VIST. My question was considering this scenario: if the pretraining did not involve the temporal ordering task, would that affect VIST as a downstream task significantly? Essentially, what is the importance of temporal ordering during pretraining as compared to only fine-tuning? I agree with the response that a 'zero-shot' setting would be complicated.


**Time Spent Reviewing:**

4

---

> ### Author Response · Authors · 2021-08-10
> **Author response to Reviewer CWuA**
>
> Thanks for your thoughtful review!
>
> ### Ablations for: What’s the impact of our pre-processing steps? [1a / 1b]
>
> While it's prohibitive to ablate each step of our preprocessing pipeline (MERLOT takes several days to pretrain on a v3-1024 TPU, which we have only academic level access to) we agree it would be valuable to better understand the impact of key dataset curation decisions. Based on your comment, we trained MERLOT from scratch under two new settings:
>
> 1. Ablate YT-Temporal-180M vs HowTo100M, controlling for dataset size. Instead of pretraining on the whole set, we trained on an IID sample of YT-Temporal-180M that is the same size as HowTo100M.
>
> 2. Ablate denoised ASR versus raw ASR. Instead of using ASR tokens that were denoised by a separate neural network as in the main experiments, we pretrained directly using the output of YouTube's ASR systems.
>
> Results (after the same number of iterations as 8 YTT-180M epochs, same as the paper) are as follows, on VCR Q->A validation accuracy:
>
>
> | *Model*                                    |        *VCR Q->A*     |
> |--------------------------------------------|-----------------------|
> |MERLOT (our main model on YTT180M)          |           75.15%      |
> |MERLOT (trained on HowTo100M only)          |           66.39%      |
> |Extra Ablation 1 (HowTo100M-sized YTT180M)  |           72.88%      |
> |Extra Ablation 2 (raw ASR)                  |           72.83%      |
>
> These results suggest that 1) diverse pretraining data helps (even controlling for size) and 2) that cleaning the ASR was useful. We are happy to add these results to a potential camera ready, and thank the reviewer for the suggestion.
>
> One additional clarifying note (that might address [1b]) -- we applied the same preprocessing steps for all video IDs. In other words, our “HowTo100M” data refers to our pipeline (including filtering and denoising) over video IDs from HowTo100M. Roughly 20% of these videos were discarded, leaving us with 800k videos; the “smaller-YT-Temporal-180M” ablation accordingly also uses 800k filtered videos. This suggests that the preprocessing steps are already baked into the HowTo100M numbers in Table 3B. We’ll clarify this in revision.
>
> ### How does MERLOT perform on VIST without the temporal ordering objective? [1c]
>
> Thanks for the great question! We believe that our presentation in the submitted version may have caused a slight confusion. Our frame-reordering pretraining task gives MERLOT an interface that enables direct decoding of its predictions for VIST. By using this objective, our model explicitly predicts which frame goes where temporally, allowing for direct decoding. We are not sure of how to do VIST in a “zero-shot” manner without this interface, unless we make independence assumptions between vision and language (like what was used for the CLIP baseline). We will improve the clarity of this in revision.
>
> Another option would be to finetune the model, but this would give it an additional advantage over other ‘zero-shot’ models. If you have other ideas for experiments you’d like us to run on this corpus, we would be happy to add them into the camera ready.
>
> ### More information about the dataset itself, e.g., topics, clip length, etc.? [2b]
>
> We are happy to expand appendices A and B, which describe the dataset and explore its content, in a potential camera ready. In addition to topics (e.g., cooking, legal, etc.) and types of videos (e.g., creator type, music video vs. instructional vs. news), as the reviewer suggested, we can add some additional statistics about the length of the clips. While there is some additional merging and splitting described in appendix A.4, when we sample L=32 tokens, the clips are on average roughly 12 seconds, with few exceeding 25 seconds, and almost none exceeding 50.
>
> ### Can other models use the strategy of drawing bounding boxes directly onto images (for VCR)?
>
> For tasks like VCR and TVQA+ where visual references are given as part of the input (e.g., "Person X" is referenced in the question + bounded explicitly in the image), VisualBERT-style models are generally made to incorporate this grounding directly, via a bounding box input (plus an extra learned embedding). This is because they don’t process the image directly, but rather, use bounding boxes from a supervised object detector like Faster-RCNN. For our work with MERLOT, however, we wanted to avoid using a (supervised) object detector backbone, which is why we needed to draw on bounding boxes.
>
> While our intent isn't to claim that our method is better (beyond the fact that we needed it for VCR and TVQA+), the reviewer is correct that other VisualBERT-style models could, in theory, use it. But, it's not clear exactly how to correctly apply Faster-RCNN to a colored-bounding box image: adding, e.g., a green hue to the ground-truth bounded region seems superfluous in the context of explicitly passing the reference, anyway.
>
> While a thorough comparison of methods for passing grounded inputs to V+L models is beyond the scope of our work, we also performed an ablation with MERLOT on VCR, without rendering bounding boxes to the image (instead forcing the model to automatically learn the disambiguation), which we will include in revision. For this experiment, we used the same model that we used for the original VCR results -- MERLOT trained on 40 YT-Temporal-180M epochs (with more training time than we could use for ablations).
>
> |  *Model*                                    |        *VCR Q->A validation accuracy*     |
> |--------------------------------------------|-----------------------|
> | MERLOT (with drawn-on bounding boxes)      |           79.4%      |
> | MERLOT (with no bounding boxes)            |           74.8%      |
>
>
> This 'without boxes' ablation is about on par with UNITER-base (75.0% accuracy, though on test and not val). Drawing on the boxes (and thus providing MERLOT with what UNITER is provided) improves performance by over 4 percentage points, suggesting that it is a possible way to encode the 'referring expression' information.
>
> ### Minor comments
>
> Thanks for pointing these out, we will address these in revision. For c) -- segments of the video without speech transcripts are indeed ignored.

---

### Review · Ethics_Reviewer_EpSa · 2021-08-06

**Recommendation:** I believe the concerns can be address…

**Ethical Issues:**

Yes

**Ethics Review:**

I think the paper raises some additional ethical issues that are not fully discussed, that I believe can be addressed:
i) With respect to privacy, I think there may be an issue with using public data from the YouTube API. I am not sure that the videos being public means that the youtubers in question are okay with this data being used for research. Even though the videos are public, it is not clear that the youtubers anticipated that they would be used as part of a dataset; in turn, their data might reach a much larger audience than they expected (subscribers and youtube users getting the youtubers videos as recommendations). It would be nice to discuss this more and to elaborate on the distinction between a) publicly available data being used and b) youtubers consenting to their data being used/providing their data themselves.
ii) It would be nice to add some discussion of the negative ways in which MERLOT (nice name!) could be used. I think the ability to perform large scale "temporal common sense" understanding can be used to automatically/systematically target some populations in possibly icky way: for example it could lead to applications that systematically target political opponents based on footage of them opposing the governement It is also not impossible that it could be used to systematically infer private/sensitive information about individuals based on their behavior. It would be nice to see more discussion of such possible malicious uses of such a technology.

---

> ### Author Response · Authors · 2021-08-18
> **Author response to Reviewer EpSa**
>
> Thanks for the thoughtful review, and the helpful suggestions! We’ll use them to improve the discussion in the paper.
>
> ###  Privacy -- using public data through the YouTube API
>
> Thanks for the note -- we discussed a similar point also in our response to Reviewer SyKv under "Fair use and possible consent harms." In short, users consent, through agreeing to the YouTube terms of service, for others to access their videos through YouTube (which by our interpretation includes research-based use). We also took a few steps to go beyond this standard, like tailoring our data curation strategy to prioritize celebrities, and by not sharing the videos ourselves -- thereby giving users the “right to be forgotten” by the Internet entirely by deleting their videos if they so choose. We hope that our proactive steps here catch on as generally accepted norms of the community.
>
> ### Discussion of possible uses for negative cases
>
> We agree that future technology for temporal commonsense understanding, like most ML/AI technologies, risk dual use. In revision, we’ll better discuss these types of malicious, largely surveillance-based use-cases, that relate to not just this work but the fields of machine learning, computer vision, and NLP more broadly.
>
> In addition to highlighting possible negative impacts, we also hope to deter them. Indeed, this is part of why we feel that MERLOT is important to release now (for research use) -- doing so can give researchers a head start in studying these negative use cases, and creating defenses against them, well before they become commonplace (if that indeed happens). We also hope that greater study can inform researchers, the general public, and policymakers about possible risks.
>
> At a lower level, we did some preliminary experiments about our model’s performance on zero-shot tasks (which we suspect might be the riskiest, in terms of dual-use by low-skilled actors). Though the model is able to order video frames, we found that its masked-language modeling interface is not very good at things like associating individuals with their name, or even correctly answering some types of visual questions out-of-the-box (like “what color is the shirt?”). It picks up new capabilities when finetuning (e.g. on VCR), but in a small-data setting, the model frequently makes simple errors. This is different from models like GPT3, which perform quite well at few- and zero-shot language completion tasks, and thus might have greater dual-use risk in this regard. Some reasons for this might be that MERLOT has 100x fewer parameters as GPT3, and was trained on noisy data (machine-generated YouTube transcripts).
>
> Overall, the preliminary results make us feel okay releasing it for explicitly non-commercial, research use. We hope to continue, along with the community, of studying models like MERLOT to probe the kinds of knowledge that they contain, which can help inform a proactive response towards dual use concerns of multimodal models.

---

### Review · Ethics_Reviewer_SyKv · 2021-08-11

**Recommendation:**

The authors do an especially good job of highlighting the potential ethical concerns and negative societal impacts of their work. But their suggestions for mitigating these problems are weak. On one hand, none of the issues are not unique to this project—they are endemic to field-wide practices of training machine learning models on public datasets, of developing models on single-language corpora, and of relying on high energy consumption training methods. On the other hand, merely acknowledging the problems does not do much to help resolve them. It would be useful to hear more from the authors about why the benefits of this work in particular outweigh the potential harms they recognize, and about how they might help the field as a whole address them, rather than perpetuate them.

**Ethical Issues:**

Yes

**Ethics Review:**

The paper describes training a model on millions of YouTube videos, raising privacy concerns (information disclosed in one context being put to use in another) and about lack of consent from video publishers/subjects. Given that the model is trained exclusively on English-language videos, its performance is likely degraded for non-English content. Model training is energy intensive.

---

> ### Author Response · Authors · 2021-08-18
> **Author response to Reviewer SyKv**
>
> Thanks for the helpful and detailed review! We appreciate the remark that we did “an especially good job of highlighting the potential ethical concerns and possible societal impacts.” We agree that the issues related to this line of research are “endemic to field-wide practices”; at the same time, we hope to use this work to better highlight, and minimize, harms related to these practices.
>
> ### What do we mean by “in public without being public”? (from Marwick and boyd 2011)
>
> Thanks for pointing out the ambiguity here, we’ll clarify it in revision. We agree with your interpretation -- aggregating (and not distributing) videos from many sources is different than, say, training a model on videos from a single channel (or person). We acknowledge that it isn’t a complete solution.
>
> ### Fair use and possible consent harms?
>
> We agree that respecting users, and being aware of their expectations of privacy, is important -- not just to our work but to the entire field. This perspective informed our data-curation strategy, and we hope further that by adding onto the discussion we currently have in the paper, we can continue this dialogue at a community-wide level.
>
> With respect to YouTube specifically, we consulted with a lawyer while doing this work, and came to the understanding that our use of videos on the platform can be considered “fair use”. By posting content to YouTube, users have to agree to its Terms of Service. The terms of this service stipulate that, when a video is uploaded to YouTube as “public”, other users have the right to access it.
>
> Prior work, in particular [Kang et al. 2015’s study](https://www.usenix.org/system/files/conference/soups2015/soups15-paper-kang.pdf), suggests that YouTubers generally understand the terms of this policy (as pertaining to privacy). This differentiates YouTube from more private venues, like email and social media (e.g., N07 thought that YouTube is open to “a lot of other people," C06: “I think there’s a user profile [on YouTube]. I mean that to me is a much more public space.”). This is possibly because "social cues on sites like YouTube... e.g., number of views... indicated the presence of other users, which rendered participants’ activity on those sites more public." This suggests that users understand the privacy implications of the platform, going beyond agreeing to its terms of service.
>
> In addition to expanding this discussion in the paper in hopes of continuing this dialogue at a community level, we did take several concrete, proactive steps beyond legal standards of privacy/copyright. We:
> 1. do not release videos for download; which gives users an explicit “right to be forgotten” from not just YouTube, but our data as well (see our response to w8dn),
> 2. tailor our data curation strategy towards channels with many subscribers/celebrities/etc. (and thus our use of their likenesses can be considered journalistic)
> 3. performed analysis of this data curation strategy, which we present in the appendix; showing that it was broadly effective at filtering out smaller YouTube channels.
>
> We hope that our data release strategy and the discussion prompted by the ethics reviewers can help advance norms for the community going forward; we would appreciate any additional feedback or suggestions for how to improve. We’ll also discuss ongoing related work that can help preserve various notions of user privacy (see the comment under “How can we help the field as a whole address, not perpetuate, harms?”)
>
> ### Not advocating for deployed use cases
>
> Thanks for pointing out the ambiguity! We’ll clarify what we mean by this in revision: we mean that, while we hope that our model can be used by the community for *research*, we actively do not want the model to be used for production purposes (similar stipulations were made by the authors of CLIP in their model card: "Any deployed use case of the model - whether commercial or not - is currently out of scope." https://github.com/openai/CLIP/blob/main/model-card.md).
>
> We see research accessibility as going hand-in-hand with saving energy through amortization. For example, there has been a lot of work in the research community at identifying social biases of models, and pointing out dual-use issues (along with defenses against them). Releasing MERLOT for research purposes thus helps academic researchers to do this critical work without the carbon expense of retraining an entire model.
>
> It is possible, however, that our policy of prohibiting (to the best of our ability) the use of our MERLOT model in production, might encourage a company to re-train an identical model for those same production purposes. We accept this tradeoff and think that giving other researchers a head-start on analyzing our model outweighs the possible carbon cost. We think it is important to enable the community to comprehensively study harms related to social bias and dual-use, before a possible time in which a model like MERLOT is placed in production. If the paper is accepted, we hope to also contribute to these follow-up areas.
>
>
> ### How can we help the field as a whole address, not perpetuate, harms?
>
> Thanks again for the detailed comments. Overall, we hope that we can help the field address its endemic harms through a few ways, of which we’ve discussed a few:
> Setting higher standards than what is currently accepted for public datasets, in terms of both consent and privacy, through our release strategy
> Amortizing the cost of training large models
> Studying issues of bias, dual-use, etc. of video-based pretraining, and giving researchers the tools to meaningfully continue this research
> Making training and inference more carbon-efficient (for example, versus other detection-based approaches)
>
> At a higher level, we view this work as being a part of a larger conversation on the limits of pretrained “foundation models.” In revision, we will discuss further, particularly with respect to the recent survey from [Bommasani et al 2021](https://arxiv.org/abs/2108.07258), about these models and their societal impacts. These span many areas of society, including healthcare, law, and education, and at the same time have significant risks including fairness, bias, energy, and dual use (as Reviewer EpSa points out as well).
>
> We feel that studying this modeling paradigm is important, particularly in the realm of academic research, instead of (premature) commercial use. We believe that this conversation needs to be grounded in technical work, much of which doesn’t exist (yet) in the video-and-language space. For instance, related to privacy, to what extent is training data encoded in the model weights? Does this leak (private) information? A paper by [Carlini et al, 2021](https://arxiv.org/abs/2012.07805) investigates this in the context of language models, but it is unclear right now what it would look like in multimodal video models. We hope that  dissemination of our work enables the community to study this further, which we hope to contribute to as well, along with the parallel question of using different methodology (e.g. from differential privacy) to better preserve privacy, should it be leaked.
>
> We appreciate any feedback for how we (and the community at large) can better perform this research.

---

### Author Response · Authors · 2021-08-10
**Author response -- overview**

We thank the reviewers for the helpful comments. We appreciate encouraging remarks that our paper is a “strong submission” (CWuA) and a “good paper” (aTvC) in presenting an approach for learning “powerful multimodal representations from videos and transcripts using no labeled data” (eAHy). Through training on videos instead of images (eAHy), and “interesting innovations on the model side” (aTvC), our model MERLOT achieves SOTA on a variety of video/vision-language understanding tasks (CWuA, w8dn, aTvC, eAHy).

Reviewers said that the experiments and ablation studies were extensive (eAHy) and well-executed (w8dn). The reviewers asked for a few additional ablations which we provided in comment responses; we plan to include and discuss these in revision which should strengthen the paper even more.

Reviewers noted that releasing the YT-Temporal-180M dataset will be impactful for the community (w8dn, aTvC). At the same time, our aim with this work was to release the data carefully, while maximally preserving user privacy. We discussed our release strategy in the paper (Section 6), which was commented on favorably by aTvC, eAHy, and CWuA; we include additional discussion in our response to w8dn.

---

### Decision · Program_Chairs · 2021-09-27

**Decision:**

Accept (Oral)

**Comment:**

This work proposes to jointly learn a visual-language model from a large amount of uncurated videos for (YT-Temporal-180M) with transcribed speech. The authors combine a set of pretraining tasks: image-text contrastive learning, MLM (with a special twist of using attention for selecting words), and temporal predictions of video frames. Reviewers unanimously this is good paper with strong SOTA results across many benchmarks and thoughtful analyses. The authors also thoughtfully respond to ethic reviewers' comments. On my side, I particularly appreciate the additional interesting ablation tests on data processing and ASR that the authors provided during rebuttal. The YT-Temporal-180M seems like a valuable resource for researchers and I hope that a careful release strategy of the dataset will work. I hope the authors can add more details to "Section 5.2 Qualitative analysis: MERLOT learns representations spanning distant frames" and make the figure more viewable. Last point, the title refers to "script knowledge" and to be convincing on that, it would be helpful to have a qualitative analysis on the model understanding of one scenario, e.g., restaurant in details. Given all the plus points of this paper, I recommend Accept.